# Prediction and characterization of enzymatic activities guided by sequence similarity and genome neighborhood networks

Suwen Zhao[1†], Ayano Sakai[2†], Xinshuai Zhang[2†], Matthew W Vetting[3†], Ritesh Kumar[2†], Brandan Hillerich[3], Brian San Francisco[2], Jose Solbiati[2], Adam Steves[4], Shoshana Brown[4], Eyal Akiva[4], Alan Barber[4], Ronald D Seidel[3], Patricia C Babbitt[4], Steven C Almo[3]*, John A Gerlt[2,5,6]*, Matthew P Jacobson[1]*

[1]Department of Pharmaceutical Chemistry, University of California, San Francisco, San Francisco, United States; [2]Institute for Genomic Biology, University of Illinois at Urbana-Champaign, Urbana, United States; [3]Department of Biochemistry, Albert Einstein College of Medicine, New York, United States; [4]Department of Bioengineering and Therapeutic Sciences, University of California, San Francisco, San Francisco, United States; [5]Department of Biochemistry, University of Illinois at Urbana-Champaign, Urbana, United States; [6]Department of Chemistry, University of Illinois at Urbana-Champaign, Urbana, United States

*For correspondence: steve. almo@einstein.yu.edu (SCA); j-gerlt@illinois.edu (JAG); matt@ cgl.ucsf.edu (MPJ)

†These authors contributed equally to this work

**Competing interests:** The authors declare that no competing interests exist.

**Reviewing editor**: Jon Clardy, Harvard Medical School, United States

**Abstract** Metabolic pathways in eubacteria and archaea often are encoded by operons and/or gene clusters (genome neighborhoods) that provide important clues for assignment of both enzyme functions and metabolic pathways. We describe a bioinformatic approach (genome neighborhood network; GNN) that enables large scale prediction of the in vitro enzymatic activities and in vivo physiological functions (metabolic pathways) of uncharacterized enzymes in protein families. We demonstrate the utility of the GNN approach by predicting in vitro activities and in vivo functions in the proline racemase superfamily (PRS; InterPro IPR008794). The predictions were verified by measuring in vitro activities for 51 proteins in 12 families in the PRS that represent ~85% of the sequences; in vitro activities of pathway enzymes, carbon/nitrogen source phenotypes, and/or transcriptomic studies confirmed the predicted pathways. The synergistic use of sequence similarity networks[3] and GNNs will facilitate the discovery of the components of novel, uncharacterized metabolic pathways in sequenced genomes.

## Introduction

The explosion in the number of sequenced eubacterial and archaeal genomes provides a challenge for the biological community: >50% of the proteins/enzymes so identified have uncertain or unknown in vitro activities and in vivo physiological functions. Genome context can provide important clues for assignment of functions to individual enzymes and, also, guide the discovery of novel metabolic pathways: pathways often are encoded by operons and/or gene clusters. However, large-scale approaches are required to efficiently mine this information for entire protein/enzyme families (*Dehal et al., 2010*; *Caspi et al., 2012*; *Markowitz et al., 2012*; *Franceschini et al., 2013*; *Overbeek et al., 2014*).

In this manuscript, we describe the use of a new bioinformatic strategy, genome neighborhood networks (GNNs), to discover the enzymes, transport systems, and transcriptional regulators that constitute metabolic pathways, thereby facilitating prediction of their individual in vitro activities and combined in vivo

**eLife digest** DNA molecules are polymers in which four nucleotides—guanine, adenine, thymine, and cytosine—are arranged along a sugar backbone. The sequence of these four nucleotides along the DNA strand determines the genetic code of the organism, and can be deciphered using various genome sequencing techniques. Microbial genomes are particularly easy to sequence as they contain fewer than several million nucleotides, compared with the 3 billion or so nucleotides that are present in the human genome.

Reading a genome sequence is straight forward, but predicting the physiological functions of the proteins encoded by the genes in the sequence can be challenging. In a process called genome annotation, the function of protein is predicted by comparing the relevant gene to the genes of proteins with known functions. However, microbial genomes and proteins are hugely diverse and over 50% of the microbial genomes that have been sequenced have not yet been related to any physiological function. With thousands of microbial genomes waiting to be deciphered, large scale approaches are needed.

Zhao et al. take advantage of a particular characteristic of microbial genomes. DNA sequences that code for two proteins required for the same task tend to be closer to each other in the genome than two sequences that code for unrelated functions. Operons are an extreme example; an operon is a unit of DNA that contains several genes that are expressed as proteins at the same time.

Zhao et al. have developed a bioinformatic method called the genome neighbourhood network approach to work out the function of proteins based on their position relative to other proteins in the genome. When applied to the proline racemase superfamily (PRS), which contains enzymes with similar sequences that can catalyze three distinct chemical reactions, the new approach was able to assign a function to the majority of proteins in a public database of PRS enzymes, and also revealed new members of the PRS family. Experiments confirmed that the proteins behaved as predicted. The next challenge is to develop the genome neighbourhood network approach so that it can be applied to more complex systems.

metabolic functions. As the first demonstration of its use, we applied this approach to the functionally diverse proline racemase superfamily (PRS) and predicted functions for >85% of its members. The predictions were verified using high-throughput protein expression and purification, in vitro enzyme activity measurements, microbiology (phenotypes and transcriptomics), and X-ray crystallography.

Three enzymatic activities have been described for the PRS: proline racemase (ProR; eubacteria [*Stadtman et al., 1957*] and eukaryotes [*Reina-San-Martín et al., 2000*], 4R-hydroxyproline 2-epimerase (4HypE; eubacteria [*Adams and Frank, 1980*; *Goytia et al., 2007*; *Gavina et al., 2010*]), and *trans* 3-hydroxy-L-proline dehydratase (*t*3HypD; eukaryotes [*Visser et al., 2012*] and eubacteria [*Watanabe et al., 2014*]); these reactions and the pathways in which they participate are shown in *Figure 1*. The previously characterized ProRs and 4HypEs catalyze racemization/epimerization of the α-carbon in a 1,1-proton transfer mechanism that, in the structurally characterized enzymes, uses two general acidic/basic Cys residues located on opposite faces of the active site (*Buschiazzo et al., 2006*; *Rubinstein and Major, 2009*). The *syn*-dehydration reaction catalyzed by *t*3HypD requires a general basic catalyst to abstract the proton from the α-carbon; its conjugate acid likely functions as the general acidic catalyst to facilitate departure of the 3-hydroxyl group. Sequence alignment of the functionally characterized *t*3HypDs and ProRs suggests the presence of a single active site Cys residue in the active sites of the *t*3HypDs (the second Cys in ProR is replaced by a Thr residue).

## Results

### Sequence similarity network for the PRS

A sequence similarity network (SSN) (*Atkinson et al., 2009*) for 2333 unique sequences in the PRS (InterPro family IPR008794; release 43.0) was constructed and displayed at various e-value thresholds (*Figure 2*). When the network is displayed with an e-value threshold of $10^{-55}$ (> ~35% sequence identity is required to draw an edge [line] between nodes [proteins]), the majority of the members of the PRS are located in a single functionally heterogeneous cluster (*Figure 2A*). As the e-value threshold stringency is increased to $10^{-110}$ (sequence identity required to draw an edge is increased to > ~60%),

**Figure 1**. The reactions catalyzed by proline racemase (ProR), 4R-hydroxyproline 2-epimerase (4HypE), and *trans*-3-hydroxy-L-proline dehydratase (*t*3HypD) and the metabolic pathways in which they participate. *c*Hyp oxidase, Pyr4H2C deaminase, α-KGSA dehydrogenase, and Δ¹-Pyr2C reductase belong to the D-amino acid oxidase (DAAO), dihydrodipicolinate synthase (DHDPS), aldehyde dehydrogenase, and ornithine cyclodeaminase (OCD) (or malate/L-lactate dehydrogenase 2 [MLD2]) superfamilies, respectively. Abbreviations: L-Pro: L-proline; D-Pro: D-proline; 5-AV: 5-aminovalerate; *t*4Hyp: *trans*-4-hydroxy-L-proline; *c*4Hyp: *cis*-4-hydroxy-D-proline; Pyr4H2C: Δ¹-pyrroline 4-hydroxy 2-carboxylate; α-KGSA: α-ketoglutarate semialdehyde; α-KG: α-ketoglutarate; *t*3Hyp: *trans*-3-hyroxy-L-proline; Δ²-Pyr2C: Δ²-pyrroline 2-carboxylate; Δ¹-Pyr2C: Δ¹-pyrroline 2-carboxylate.

the PRS separates into 28 clusters and 49 singletons (*Figure 2B*). For analyses of the genome neighborhoods (vide infra), each cluster in the $10^{-110}$ network was assigned a unique color and number as shown in *Figure 2B* (the node colors in *Figure 2A* depict their association with the clusters in *Figure 2B*).

At the e-value threshold of $10^{-110}$ (*Figure 2B*) the nodes for the experimentally characterized functions—ProR (magenta; cluster 7), 4HypE (blue and red; clusters 1 and 2, respectively), and *t*3HypD (brown; cluster 8)—are located in separate clusters that account for ~30% of the sequences in the PRS. When the e-value threshold is relaxed to $10^{-55}$, most of the clusters merge, although the nodes associated with the two previously characterized 4HypE clusters in the $10^{-110}$ network remain separated. Sequence alignments predict that the active sites of both characterized 4HypE clusters contain two active site Cys residues. We conclude that these two families of 4HypEs evolved from divergent, but homologous, ancestors.

At the e-value threshold of $10^{-110}$ (*Figure 2B*), the separated clusters are expected to be isofunctional because, from sequence alignments, their active sites are formed from conserved amino acid residues (acid/base catalysts and specificity determining residues). Although many of the clusters are predicted to have the two active site Cys residues found in the structurally characterized ProR (PDB: 1W61) and 4HypE (PDB: 2AZP [*Liu et al.*]), others are missing one or both of the Cys residues. The previously uncharacterized enzymes with differing residues could either represent new functions or additional examples of evolution of the ProR, 4HypE, and *t*3HypD functions from divergent, but homologous, ancestors.

## GNN for the PRS

We predicted functions for ~80% of the remaining members of the PRS by analyzing the SSN for the proteins (including enzymes, transport systems, and transcriptional regulators) encoded by the genome

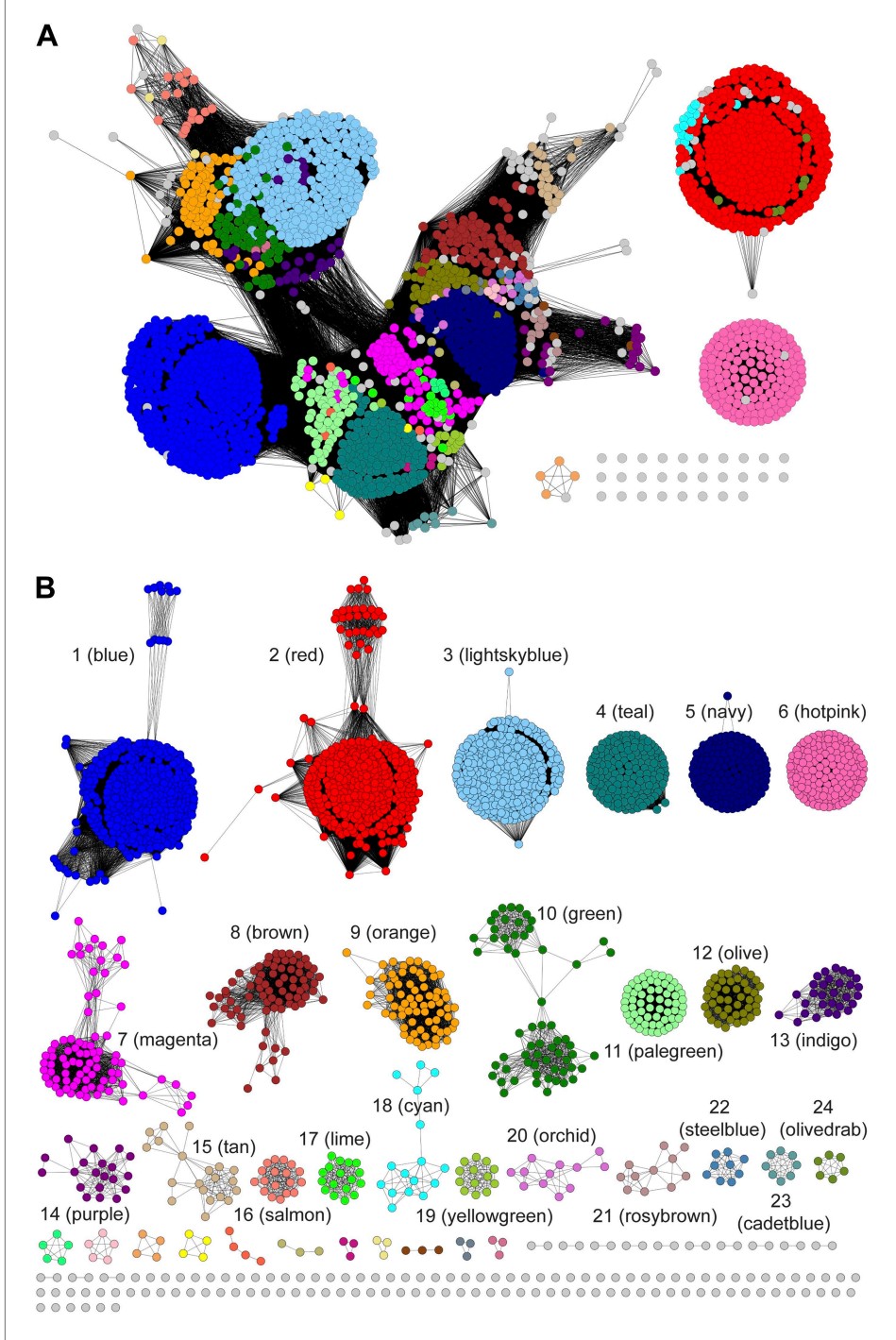

**Figure 2**. Sequence similarity networks (SSNs) for the PRS. (**A**) The SSN displayed with an e-value threshold of $10^{-55}$ (~35% sequence identity). (**B**) The SSN displayed with an e-value threshold of $10^{-110}$ (~60% sequence identity).

neighborhoods for 'all' members of the PRS (specifically, ± 10 genes relative to the gene encoding each PRS member, the query). A protein in this genome neighborhood SSN, designated the 'genome neighborhood network' (GNN), is expected to be functionally related to a query in the PRS if they are located in an operon and/or gene cluster that encodes a metabolic pathway that includes the query. By analyzing many genome neighborhoods simultaneously, e.g., for all members of the PRS, the signals associated with functionally related proteins will be amplified; the signals associated with functionally

unrelated genome proximal proteins that occur 'randomly' across many species will contribute to the background 'noise'. We propose that this large-scale approach is more efficient in identifying 'all' of the enzymes/transport systems/transcriptional regulators in a conserved metabolic pathway than by a one-genome-at-a-time analysis.

Our approach for visualizing a GNN first assigns a unique query color and number to the members of each cluster in the input SSN that separates the members of the PRS into clusters that are likely to be isofunctional (e$^{-110}$ in this work). After collecting the genome neighbors, we assign each of them the same color as the color of the query; with this strategy, proteins that are encoded by the same genome neighborhood as the query are easily identified in the GNN because they share the same color as the query. We then perform an all-by-all BLAST on the sequences of the genome neighbors and display the results as an SSN using an e-value threshold of 10$^{-20}$; this SSN is the GNN. Using this e-value threshold, most of the clusters in the GNN contain the members of distinct protein families and superfamilies (e.g., Pfam families); however, in some cases, divergent families in functionally diverse superfamilies may be found in separate clusters. Genome neighborhood proteins that occur randomly across divergent species and are functionally unrelated to the queries are expected to be located in small clusters with multiple colors, so these can be quickly identified visually and discarded from further analysis. The PRS queries from the input SSN ('zero sequences' in collecting the ±10 neighbors) are not displayed in the GNN, except when multiple members of the PRS are proximal on the genome, that is, when one PRS member is in the genome neighborhood of another (vide infra).

The GNN for the PRS (*Figure 3A*) contains many clusters (protein families). In some clusters, all of the nodes have the same color, that is, they are identified by a single query cluster in the SSN (e.g., the clusters in *Figure 3B,C*). However, in most clusters the nodes have multiple colors, that is, they are identified by several query clusters in the SSN (e.g., the clusters in *Figure 3D–H*); this suggests that different query clusters in the SSN have the same in vitro activity and in vivo metabolic function. The clusters in the GNN (*Figure 3A*) are labeled with their Pfam annotations. The ligand/substrate specificities and/or reaction mechanisms that characterize these families are then used to predict the individual in vitro activities and the shared metabolic pathway identified by a query cluster.

## Retrospective tests of GNN: ProR and 4HypE functions

As a retrospective use of the GNN, the ProR function is encoded by anaerobic eubacteria that ferment L-proline and is represented by the magenta cluster (cluster 7) in the SSN (*Figure 2B*). The first step in the catabolism of L-proline is racemization to D-proline (by ProR) that is reduced to 2-keto-5-aminopentanoate by D-proline reductase (*Kabisch et al., 1999*) (by PrdAB; *Figure 1*). In the GNN, the clusters for the PrdA and PrdB polypeptides in D-proline reductase are uniformly magenta, as expected if the genes encoding ProR and PrdAB are colocalized with the gene encoding ProR (*Figure 3B,C*). The lack of other colors in the PrdAB clusters in the GNN implies that no other clusters in the SSN have the ProR function.

As a second retrospective example, the 4HypE function has been assigned to members of the blue (cluster 1) and red (cluster 2) clusters in the SSN (*Figure 2B*). In the GNN, clusters identified by the blue and red clusters include the D-amino acid oxidase (DAAO; *Figure 3D*) (*Watanabe et al., 2012*), dihydrodipicolinate synthase (DHDPS; *Figure 3E*) (*Singh and Adams, 1965*; *Watanabe et al., 2012*), and aldehyde dehydrogenase (*Figure 3F*) (*Koo and Adams, 1974*; *Watanabe et al., 2007*) superfamilies as well as components of several types of transport systems. As we and others recently established for organisms that use *trans*-4-hydroxy-L-proline betaine as sole carbon and nitrogen source (*Zhao et al., 2013*; *Kumar et al., 2014*), the catabolic pathway for *trans*-4-hydroxy-L-proline (*t*4Hyp) (*Figure 1*) can be initiated by the epimerization of *t*4Hyp to *cis*-4-hydroxy-D-proline (*c*4Hyp) by 4HypE, followed by reactions catalyzed by *c*4Hyp oxidase (a member of the DAAO superfamily), *c*4Hyp imino acid dehydratase/deaminase (a member of the DHDPS superfamily), and α-ketoglutarate semialdehyde dehydrogenase (a member of the aldehyde dehydrogenase superfamily). Thus, the occurrence of blue and red nodes in these three clusters in the GNN is expected.

## Discovery of new families of 4HypEs

The DAAO (*Figure 3D*), DHDPS (*Figure 3E*), and aldehyde dehydrogenase (*Figure 3F*) clusters also contain nodes with other colors from the SSN (*Figure 2B*), including orange (cluster 9), pale green (cluster 11), and teal (cluster 4). Proteins from the orange and pale green clusters were purified and assayed using a library of proline derivatives (*Figure 4*). As expected, members of the orange and pale

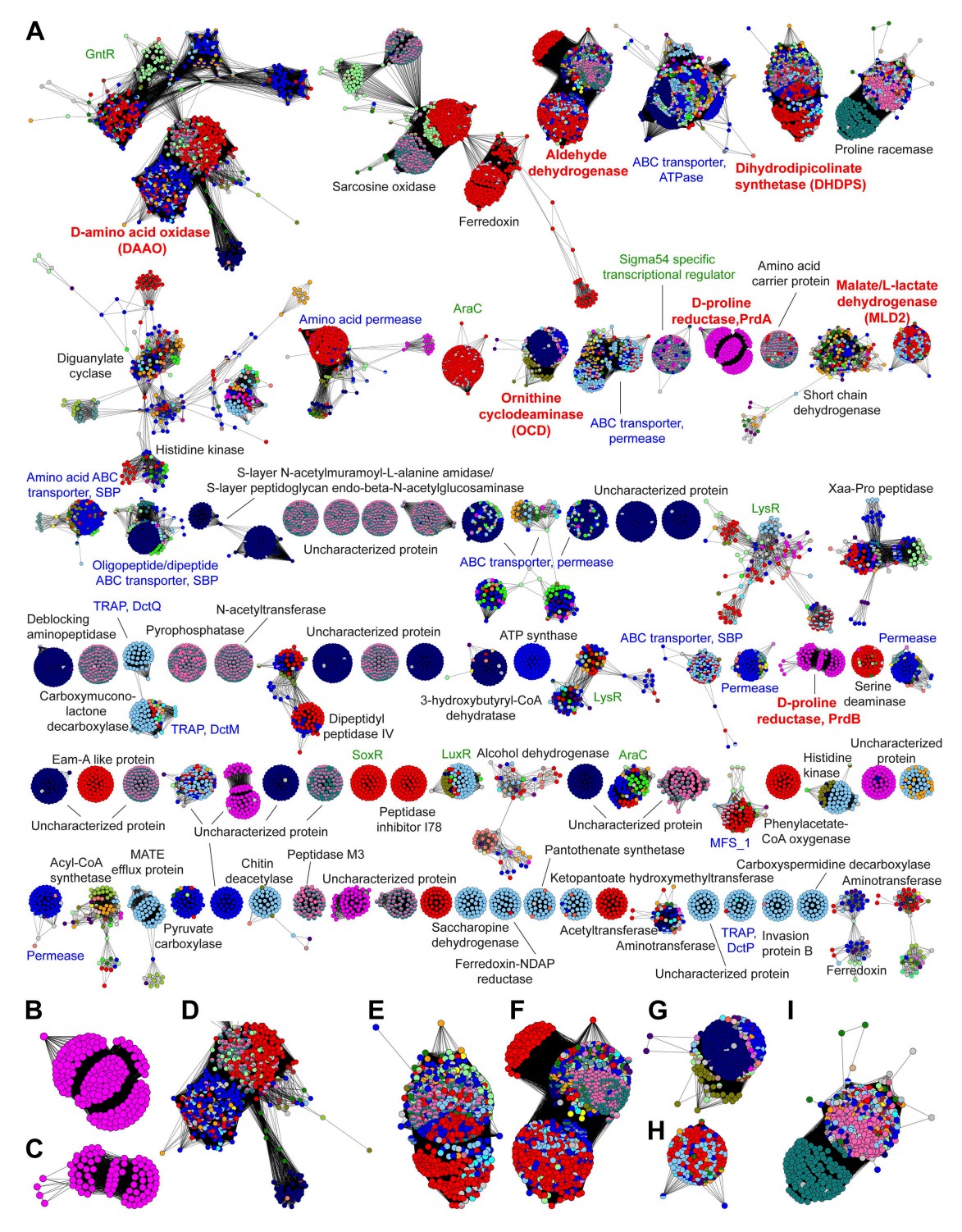

**Figure 3**. The genome neighborhood network (GGN) for the PRS. (**A**) The GNN displayed with an e-value threshold of $10^{-20}$. The nodes are colored by the color of query nodes in the SSN (*Figure 2A*). The clusters are labeled with the UniProtKB/TrEMBL annotations. (**B–I**) Selected superfamily clusters from the GNN showing node colors. (**B**) D-proline reductase PrdA. (**C**) D-proline reductase, PrdB. (**D**) D-amino acid oxidase (DAAO). (**E**) Dihydrodipicolinate synthase (DHDPS). (**F**) Aldehyde dehydrogenase. (**G**) Ornithine cyclodeaminase (OCD). (**H**) Malate/L-lactate dehydrogenase 2 (MLD2). (**I**) Proline racemase.

**Figure 4**. Library of proline and proline betaine derivatives tested for ESI-MS screening. These substrates were divided into four groups to avoid mass duplication.

green clusters catalyze the 4HypE reaction (*Tables 1 and 2*). We were unable to purify proteins from the teal cluster (insolubility), so we used the growth phenotypes of the encoding organisms and transcriptomics to identify their in vitro enzymatic activities and in vivo metabolic functions. As predicted from the GNN, *Bacillus cereus* ATCC14579 (cluster 4, teal) and *Streptomyces lividans* TK24 (cluster 11, pale green) both utilize *t*4Hyp as sole carbon source (*Table 3*); also, the genes encoding the predicted 4HypEs (*Table 4*) and the proximal genes encoding the predicted *c*4Hyp oxidases, *c*4Hyp imino acid dehydratase/deaminases, and α-ketoglutarate semialdehyde dehydrogenases (*Table 5*) are up-regulated when the encoding organism is grown on *t*4Hyp as carbon source (*Table 4*). The purified proteins from the orange groups are promiscuous for the 3HypE reaction (*Tables 1 and 2*), but their genome neighborhood context identifies their physiological functions as 4HypE.

## X-ray structure of a novel 4HypE

The X-ray structure of one of the previously functionally assigned 4HypEs (Uniprot: Q4KGU2; locus tag: PFL_1412; red, cluster 2) was determined in the presence of the substrate, *t*4Hyp and, also, pyrrole-2-carboxylate (PYC), a stable analogue of the enolate anion intermediate (*Figure 5A,B*; *Table 6*). These are the first liganded structures of a 4HypE and the first structure of a PRS with an authentic substrate. These structures corroborate the positioning of the active site Cys/Cys pair (Cys 88, Cys 236) to facilitate substrate epimerization, highlight residues specific to the coordination of the 4-hydroxyl group, and validate the hypothesis that PYC and substrate bind in a similar fashion. In addition, the X-ray structure of one of the newly functionally assigned 4HypEs (Uniprot: B9K4G4; locus tag: Avi_7022; orange, cluster 8) was determined in the presence of its substrate, *t*4Hyp. The active site contains Ser 93 on one face and Cys 255 on the opposite face (*Figure 5C*). Thus, despite the conserved ability of

**Table 1.** Mass spectroscopy screening results in $D_2O$. Hits were observed by mass shift for racemization/epimerization (+1) and dehydration ($-17$) for reactions performed

| Locus tag | UniProt | L-Pro | D-Pro | t4Hyp | c4Hyp | t3Hyp | cis-3-OH-L-Pro |
|---|---|---|---|---|---|---|---|
| **Cluster 1: blue** | | | | | | | |
| Pden_4859 | A3QFI1 | 0 | 0 | +1 | +1 | 0 | 0 |
| Shew_2363 | A9AQW9 | 0 | 0 | +1 | +1 | 0 | 0 |
| Bmul_5265 | A6WXX7 | 0 | 0 | +1 | +1 | +1 | +1 |
| Oant_1111 | D2QN44 | 0 | 0 | +1 | +1 | +1 | +1 |
| Slin_1478 | B9JHU6 | 0 | 0 | +1 | +1 | +1 | +1 |
| Arad_8151 | Q8FYS0 | 0 | 0 | +1 | +1 | 0 | 0 |
| BR1792 | A1BBM5 | 0 | 0 | +1 | +1 | +1 | +1 |
| **Cluster 2: red** | | | | | | | |
| A1S_1325 | A3M4A9 | 0 | 0 | +1 | +1 | +1 | +1 |
| Bamb_3550 | Q0B9R9 | 0 | 0 | +1 | +1 | +1 | +1 |
| BceJ2315_47180 | B4EHE6 | 0 | 0 | +1 | +1 | +1 | +1 |
| BMULJ_04062 | B3D6W2 | 0 | 0 | +1 | +1 | +1 | +1 |
| BTH_II2067 | Q2T3J4 | 0 | 0 | +1 | +1 | -17 | 0 |
| CV_2826 | Q7NU77 | 0 | 0 | +1 | +1 | +1 | +1 |
| Csal_2705 | Q1QU06 | 0 | 0 | +1 | +1 | 0 | 0 |
| PFL_1412 | A5VZY6 | 0 | 0 | +1 | +1 | +1 | +1 |
| Pput_1285 | Q1QBF3 | +1 | +1 | +1 | +1 | +1 | +1 |
| Pcryo_1219 | A3M4A9 | 0 | 0 | +1 | +1 | +1 | +1 |
| XCC2415 | Q8P833 | 0 | 0 | +1 | +1 | +1 | +1 |
| Bmul_4447 | A9AL52 | 0 | 0 | +1 | +1 | +1 | +1 |
| ABAYE2385 | B0VB44 | 0 | 0 | +1 | +1 | +1 | +1 |
| BURPS1106B_1521 | C5ZMD2 | +1 | +1 | +1 | +1 | +1 | +1 |
| BURPS1710b_A1887 | Q3JHA9 | 0 | 0 | +1 | +1 | +1 | +1 |
| PA1268 | Q9I476 | 0 | 0 | +1 | +1 | 0 | 0 |
| **Cluster 3: ligthskyblue** | | | | | | | |
| Pden_1184 | A1B195 | 0 | 0 | 0 | 0 | -17 | 0 |
| SIAM614_28502 | A0NXQ9 | 0 | 0 | 0 | 0 | -17 | 0 |
| Atu4684 | A9CH01 | 0 | 0 | +1 | +1 | -17 | 0 |
| Avi_7022 | B9K4G4 | 0 | 0 | 0 | 0 | -17 | 0 |
| Oant_0439 | A6WW16 | 0 | 0 | +1 | +1 | 0 | 0 |
| SM_b20270 | Q92WR9 | 0 | 0 | +1 | +1 | -17 | 0 |
| BMEI1586 | Q8YFD6 | 0 | 0 | +1 | +1 | +1 | +1 |
| BR0337 | Q8G2I3 | 0 | 0 | 0 | 0 | -17 | 0 |
| **Cluster 5: navy** | | | | | | | |
| BC_0905 | Q81HB1 | 0 | 0 | +1 | +1 | -17 | 0 |
| BCE_0994 | Q73CS0 | 0 | 0 | +1 | +1 | -17 | 0 |
| BT9727_0799 | Q6HMS9 | 0 | 0 | +1 | +1 | -17 | 0 |
| **Cluster 9: orange** | | | | | | | |
| Avi_0518 | B9JQV3 | 0 | 0 | +1 | +1 | +1 | +1 |
| Atu0398 | A9CKB4 | 0 | 0 | +1 | +1 | +1 | +1 |
| RHE_CH00452 | Q2KD13 | 0 | 0 | +1 | +1 | +1 | +1 |
| Arad_0731 | B9J8G8 | 0 | 0 | +1 | +1 | +1 | +1 |
| **Cluster 11: palegreen** | | | | | | | |
| Sros_6004 | D2AV87 | 0 | 0 | +1 | +1 | 0 | 0 |
| **Cluster 12: olive** | | | | | | | |
| Bamb_3769 | Q0B950 | 0 | 0 | 0 | 0 | -17 | 0 |
| Bmul_4260 | A9AKG8 | 0 | 0 | +1 | +1 | +1 | +1 |
| **Cluster 16: salmon** | | | | | | | |
| Csal_2339 | Q1QV19 | 0 | 0 | +1 | +1 | 0 | 0 |
| Maqu_2141 | A1U2K1 | 0 | 0 | 0 | 0 | 0 | 0 |
| **Cluster 17: lime** | | | | | | | |
| Rsph17029_3164 | A3PPJ8 | 0 | 0 | +1 | +1 | 0 | 0 |
| RSP_3519 | Q3IWG2 | 0 | 0 | +1 | +1 | 0 | 0 |
| **Cluster 18: cyan** | | | | | | | |
| SIAM614_28492 | A0NXQ7 | 0 | 0 | +1 | +1 | 0 | 0 |
| SADFL11_2813 | B9R4E3 | 0 | 0 | +1 | +1 | 0 | 0 |
| SPOA0266 | Q5LKW3 | 0 | 0 | +1 | +1 | +1 | +1 |
| **Cluster 22: steelblue** | | | | | | | |
| Spea_1705 | A8H392 | 0 | 0 | 0 | 0 | -17 | 0 |
| Swoo_2821 | B1KJ76 | 0 | 0 | +1 | +1 | -17 | 0 |
| **Cluster 61:** | | | | | | | |
| Plim_2713 | D5SQS4 | 0 | 0 | +1 | +1 | +1 | +1 |

this enzyme to catalyze the 4HypE reaction (a two-base 1,1-proton transfer reaction), the Cys–Cys general acid/base pair observed in the structure of Q4KGU2 from the red cluster is not conserved. This observation highlights the structural diversity associated with evolution of function in the PRS. Without the information provided by the GNNs, the 4HypE function would not have been expected.

**Table 2.** Kinetic constants for 3/4HypE and *t*3HypD activities of the screened PRS targets

| Cluster | Locus tag | UniProt | Function | $k_{cat}$ [s$^{-1}$] | K$_M$ [mM] | $k_{cat}$/K$_M$[M$^{-1}$s$^{-1}$] |
|---|---|---|---|---|---|---|
| 1 | Pden_4859 | A1BBM5 | 4HypE | 16 ± 2 | 25 ± 5 | 630 |
| | Shew_2363 | A3QFI1 | 4HypE | 50 ± 6 | 12 ± 3 | 4000 |
| | Bmul_5265 | A9AQW9 | 3HypE | 0.34 ± 0.03 | - [a] | - [a] |
| | | | 4HypE | 5.6 ± 0.5 | 11 ± 2 | 530 |
| | Oant_1111 | A6WXX7 | 3HypE | 2.4 ± 0.2 | 31 ± 7 | 77 |
| | | | 4HypE | 89 ± 2 | 7.1 ± 0.6 | 13000 |
| 2 | BTH_II2067 | Q2T3J4 | *t*3HypD | 17 ± 3 | 26 ± 9 | 660 |
| | | | 4HypE | 40 ± 4 | 1.4 ± 0.4 | 28000 |
| | CV_2826 | Q7NU77 | 3HypE | 30± 0.6 | 57 ± 4 | 520 |
| | | | 4HypE | 70 ± 7 | 6.8 ± 3 | 10000 |
| | Pput_1285 | A5VZY6 | 3HypE | 4.8 ± 0.6 | 19 ± 5 | 250 |
| | | | 4HypE | 26 ±0.7 | 0.54 ± 0.08 | 48000 |
| | | | ProR | 2.8 ± 0.1 | 200 ± 20 | 14 |
| | XCC2415 | Q8P833 | 4HypE | 28 ± 0.4 | 0.67 ± 0.05 | 42000 |
| | | | 3HypE | 1.3 ± 0.07 | 15 ± 3 | 86 |
| 3 | Pden_1184 | A1B195 | *t*3HypD | nd [b] | nd [b] | nd [b] |
| | SIAM614_28502 | A0NXQ9 | *t*3HypD | 15 ± 0.9 | 7.8 ± 1 | 1900 |
| | Atu4684 | A9CH01 | *t*3HypD | 27 ± 1 | 4.2 ± 0.8 | 6300 |
| | | | 4HypE | 0.40 ± 0.02 | 2.0 ± 0.3 | 200 |
| | Avi_7022 | B9K4G4 | *t*3HypD | 4.3 ± 0.4 | 15 ± 3 | 280 |
| | Oant_0439 | A6WW16 | 4HypE | 0.064 ± 0.002 | 1.3 ± 0.2 | 49 |
| | SM_b20270 | Q92WR9 | *t*3HypD | 7.9 ± 0.2 | 3.8 ± 0.4 | 2100 |
| | | | 4HypE | 0.089 ± 0.01 | 6.3 ± 2 | 14 |
| | BMEI1586 | D0B556 | 3HypE | 0.085 ± 0.003 | 2.6 ± 0.4 | 33 |
| | | | 4HypE | 0.082 ± 0.005 | 4.5 ± 1 | 18 |
| | BR0337 | Q8G2I3 | *t*3HypD | 17 ± 2 | 5.1 ± 2 | 3300 |
| 5 | BCE_0994 | Q73CS0 | *t*3HypD | nd [b] | nd [b] | nd [b] |
| | | | 4HypE | 1.2 ± 0.03 | 3.2 ± 0.3 | 370 |
| | BT9727_0799 | Q6HMS9 | *t*3HypD | 23 ± 5 | 7.5 ± 3 | 3100 |
| | | | 4HypE | 0.16 | - [a] | - [a] |
| 9 | Avi_0518 | B9JQV3 | 3HypE | 0.75 ± 0.04 | 4.8 ± 0.9 | 160 |
| | | | 4HypE | 1.3 ± 0.07 | 5.6 ± 0.5 | 230 |
| | Atu0398 | A9CKB4 | 3HypE | 4.0 ± 0.6 | 25 ± 7 | 160 |
| | | | 4HypE | 0.86 ± 0.1 | 4.6 ± 2 | 190 |
| | RHE_CH00452 | Q2KD13 | 3HypE | 0.94 ± 0.06 | 2.1 ± 0.7 | 450 |
| | | | 4HypE | 1.9 ± 0.08 | 2.1 ± 0.3 | 880 |

*Table 2. Continued on next page*

*Table 2. Continued*

| Cluster | Locus tag | UniProt | Function | $k_{cat}$ [s$^{-1}$] | K$_M$ [mM] | $k_{cat}$/K$_M$[M$^{-1}$s$^{-1}$] |
|---------|-----------|---------|----------|---------------------|------------|-----------------------------------|
| 11 | Sros_6004 | D2AV87 | 4HypE | 14 ± 0.8 | 7.8 ± 1 | 1800 |
| 12 | Bamb_3769 | Q0B950 | t3HypD | 43 ± 4 | 13 ± 3 | 3400 |
| 12 | Bmul_4260 | A9AKG8 | 3HypE | 30 ± 1 | 18 ± 2 | 1700 |
| 12 | Bmul_4260 | A9AKG8 | 4HypE | 1.3 ± 0.04 | 2.7 ± 0.3 | 470 |
| 16 | Csal_2339 | Q1QV19 | 4HypE | 0.070 ± 0.005 | 2.5 ± 0.7 | 28 |
| 17 | RSP_3519 | Q3IWG2 | 4HypE | nd [b] | nd [b] | nd [b] |
| 17 | Rsph17029_3164 | A3PPJ8 | 4HypE | nd [b] | nd [b] | nd [b] |
| 18 | SIAM614_28492 | A0NXQ7 | 4HypE | 55 ± 3 | 3.2 ± 0.5 | 17000 |
| 18 | SADFL11_2813 | B9R4E3 | 4HypE | 67 ± 5 | 4.1 ± 0.8 | 16000 |
| 22 | Spea_1705 | A8H392 | t3HypD | 0.15 ± 0.03 | - [b] | - [b] |
| 22 | Swoo_2821 | B1KJ76 | t3HypD | 4.1 ± 0.4 | 6.7 ± 2 | 600 |

[a] The reaction is to slow to measure K$_m$.

[b] The reaction is slow to measure kinetic parameters.

## Discovery of novel families of *t3HypDs* and Δ$^1$-Pyr2C reductases

The *t3HypD* function previously was assigned to eukaryotic members of the PRS (*Visser et al., 2012*), so their genome neighbors are not represented in the GNN. However, the members of the navy cluster (cluster 5; species of Bacilli) identify several clusters in the GNN, including families of the components of TRAP and ABC transport systems, families of peptidases, and a family in the ornithine cyclodeaminase superfamily (OCDS); several members of the olive cluster (cluster 12) also identify the same OCDS cluster (*Figure 3G*). Members of the OCDS catalyze NAD(P)$^+$/NAD(P)H-dependent reactions that involve the ketimines obtained by oxidation of α-amino acids (*Goodman et al., 2004*; *Schröder et al., 2004*; *Gatto et al., 2006*); some have been reported to catalyze the reduction of the ketimine of proline (*Hallen et al., 2011*) (and oxidation of L-proline; *Figure 6A*). Using purified proteins, we determined that members of both the navy (cluster 5) and olive (cluster 12) clusters in the SSN

**Table 3.** Growth phenotypes of bacterial strains when grown on the indicated carbon sources

| Organism | *t4Hyp* | *c4Hyp* | *t3Hyp* | *cis*-3-OH-L-proline | L-Pro | D-glucose |
|----------|---------|---------|---------|----------------------|-------|-----------|
| *Agrobacterium tumefaciens* C58 | ++ | ++ | + | − | +++ | +++ |
| *Sinorhizobium meliloti* 1021 | ++ | ++ | + | − | +++ | +++ |
| *Labrenzia aggregate* IAM12614 | + | + | + | + | +++ | +++ |
| *Pseudomonas aeruginosa* PAO1 | ++ | ++ | + | − | +++ | +++ |
| *Paracoccus denitrificans* PD1222 | +++ | +++ | + | + | +++ | +++ |
| *Rhodobacter sphaeroides* 2.4.1 | + | + | − | − | +++ | +++ |
| *Rhodobacter sphaeroides* 2.4.1ΔRSP3519 | − | + | − | − | +++ | +++ |
| *Bacillus cereus* ATCC14579 | ++ | ++ | + | + | +++ | +++ |
| *Roseovarius nubinhibens* ISM | ++ | ++ | + | − | +++ | +++ |
| *Escherichia coli* MG1655 | − | − | − | − | +++ | +++ |
| *Streptomyces lividans* TK24 | +++ | ++ | + | ND | +++ | +++ |

'+++' represents robust growth (like growth on D-glucose); ++/+ represents slow growth phenotype; '−−' represents growth-deficient phenotype; 'ND', not determined

**Table 4.** Transcriptional analysis of PRS members

| Organism/Locus Tag | *t*4Hyp | *t*3Hyp |
|---|---|---|
| *Agrobacterium tumefaciens* C58 | | |
| A9CKB4 | 12 ± 2 | 11±1.5 |
| A9CFV0 | 3 ± 1 | NC |
| A9CH01 | 64 ± 5 | 32 ± 4 |
| *Sinorhizobium meliloti* 1021 | | |
| Q92WS1 | 5 ± 1 | 3 ± 1 |
| Q92WR9 | 5.5 ± 1.5 | 3.5 ± 1 |
| *Labrenzia aggregate* IAM12614 | | |
| A0NXQ7 | 22 ± 2 | 5 ± 1 |
| A0NXQ9 | 12 ± 2 | 6 ± 2 |
| *Pseudomonas aeruginosa* PAO1 | | |
| Q9I489 | 8 ± 2 | 5 ± 1 |
| Q9I476 | 35 ± 3 | 7 ± 2 |
| *Paracoccus denitrificans* PD1222 | | |
| A1B0W2 | 2.0 ± 0.5 | NC |
| A1B195 | NC | NC |
| A1B7P4 | NC | NC |
| A1BBM5 | 4.5 ± 0.5 | NC |
| *Rhodobacter sphaeroides* 2.4.1 | | |
| Q3IWG2 | 10 ± 1 | NC |
| *Bacillus cereus* ATCC14579 | | |
| Q81HB1 | 4 ± 1 | 4.5 ± 1 |
| Q81CD7 | 22 ± 2 | 18 ± 3 |
| *Roseovarius nubinhibens* ISM | | |
| A3SLP2 | 12 ± 2 | 4 ± 1.5 |

Fold change in expression for each gene when grown on the indicated carbon source, relative to growth on D-glucose. The identities of the bacterial species and the protein encoded by each gene are indicated. Fold-changes are the averages of five biological replicates with standard deviation (p value <0.005). NC, no change

catalyze the *t*3HypD reaction (**Tables 1 and 2**). We also determined that members of the OCDS cluster catalyze the NADPH-dependent reduction of the ketimine of proline to form L-proline (**Figure 6A,B**). The catabolic pathway for *trans*-3-hydroxy-L-proline is known to proceed by dehydration, nonenzymatic tautomerization of the dehydration product to the ketimine of proline and, finally, reduction of the ketimine to form L-proline (**Figure 1**). In the OCDS SSN (**Figure 6A**), the previously characterized proline ketimine reductases are located in clusters/families distinct from the members of the OCDS identified in our GNN. Thus, assignment of the *t*3HypD function to the members of navy and olive clusters in the SSN would not have been possible without the synergistic information contained in the GNN.

## Structure of a novel t3HypD

We determined the structure of a *t*3HypD (B9K4G4) from the light sky blue cluster (cluster 3) in the presence of PYC (**Table 6**). Instead of the typical PRS Cys/Cys pair, B9K4G4 contains Ser 90 in a similar conformation as was determined for B9JQV3 from the orange cluster (4HypE activity) and Thr 256 on the opposing face (**Figure 5D**). Thr 256 mimics the conformation of the typical PRS Cys residue but with the side-chain methylene positioned against the anomeric carbon. Again, the assignment of function enabled by the GNNs identifies convergent evolution of function within the PRS.

**Table 5.** Transcriptional analysis of genome neighborhoods

| Organism/ Locus tag | Uniprot | Enzyme | Cluster | *t*4Hyp | *t*3Hyp | L-Pro |
|---|---|---|---|---|---|---|
| *Bacillus cereus* ATCC 14579 | | | | | | |
| Bc_0905 | Q81HB1 | ProR | navy | 121 ± 11 | 87 ± 10 | NC |
| Bc_0906 | Q81HB0 | OCD | | 20 ± 3 | 14 ± 2 | NC |
| | | | | | | |
| Bc_2832 | Q81CE0 | ALDH | | 630 ± 39 | 625 ± 57 | 13 ± 2 |
| Bc_2833 | Q81CD9 | DHDPS | | 644 ± 61 | 498 ± 37 | 6 ± 0.7 |
| Bc_2834 | Q81CD8 | ProR | hot pink | 594 ± 27 | 485 ± 29 | 8 ± 1 |
| Bc_2835 | Q81CD7 | ProR | teal | 408 ± 15 | 567 ± 33 | 5 ± 0.5 |
| Bc_2836 | Q81CD6 | oxidase | | 623 ± 37 | 633 ± 42 | 10 ± 0.6 |
| | | | | | | |
| *Streptomyces lividans* TK24 | | | | | | |
| SSPG_01342 | D6EJL0 | DAAO | | 81 ± 5 | 20 ± 5 | NC |
| SSPG_01341 | D6EJK9 | oxidase | | 65 ± 9 | 6 ± 0.2 | NC |
| SSPG_01340 | D6EJK8 | oxidase | | 225 ± 22 | 30 ± 3 | 3 ± 0.4 |
| SSPG_01339 | D6EJK7 | DHDPS | | 136 ± 5 | 16 ± 0.2 | NC |
| SSPG_01338 | D6EJK6 | ProR | pale green | 171 ± 8 | 23 ± 1 | 3 ± 0.2 |
| | | | | | | |
| *Agrobacterium tumefaciens* C58 | | | | | | |
| Atu_0398 | A9CKB4 | ProR | orange | 14 ± 0.4 | 16 ± 0.6 | NC |
| | | | | | | |
| Atu_3947 | Q7CTP1 | DAAO | | NC | 4 ± 0.2 | NC |
| Atu_3948 | Q7CTP2 | AlaR | | NC | NC | NC |
| Atu_3949 | Q7CTP3 | OCD | | NC | NC | NC |
| Atu_3950 | Q7CTP4 | ALDH | | NC | NC | NC |
| Atu_3951 | A9CFU8 | LysR | | NC | NC | NC |
| Atu_3952 | A9CFU9 | DAAO | | NC | NC | NC |
| Atu_3953 | Q7CFV0 | ProR | blue | NC | NC | NC |
| Atu_3958 | Q7CTQ2 | DAAO | | NC | NC | NC |
| Atu_3959 | Q7CTQ3 | ALDH | | NC | NC | NC |
| Atu_3960 | A9CFV4 | DHDPS | | NC | NC | NC |
| Atu_3961 | Q7CTQ5 | GntR | | NC | NC | NC |
| Atu_3985 | A9CFW8 | ProC | | NC | NC | NC |
| | | | | | | |
| Atu_4675 | A9CGZ4 | DHDPS | | 148 ± 2 | 87 ± 7 | NC |
| Atu_4676 | Q7CVK1 | MLD2 | | 30 ± 5 | 40 ± 7 | NC |
| Atu_4678 | A9CGZ5 | SBP | | 198 ± 18 | 79 ± 8 | NC |
| Atu_4682 | A9CGZ9 | DAAO | | 294 ± 15 | 14 ± 3 | NC |
| Atu_4684 | A9CH01 | ProR | light sky blue | 116 ± 14 | 8 ± 1 | NC |
| Atu_4691 | A9CH04 | 2-Hacid_dh | | NC | NC | NC |

Fold changes in expression for the indicated gene when grown on the indicated carbon source, relative to growth on Dglucose. Fold changes are the averages of three biological replicates with standard deviation. NC, no change

## Discovery of additional families of 4HypEs, *t*3HypDs, and Δ¹-Pyr2C reductases

Members of the light sky blue (cluster 3) cluster in the SSN identify the same (super)families identified by both the 4HypE and *t*3HypD clusters (transport systems, transcriptional regulators, DAAO [*Figure 3D*], DHDPS [*Figure 3E*], aldehyde dehydrogenase [*Figure 3F*], and OCD [*Figure 3G*]); however, several members of the light sky blue cluster identify a GNN cluster annotated as the malate/L-lactate dehydrogenase 2 superfamily (MLD2; NADH-dependent oxidoreductases) (*Muramatsu et al., 2005*) (*Figure 3H*). Using purified members of the PRS, we determined that the light sky blue cluster is functionally heterogeneous (and some members are promiscuous) for the 4HypE and *t*3HypD functions (*Tables 1 and 2*). We also determined that members of the MLD2 superfamily in the GNN catalyze the reduction of

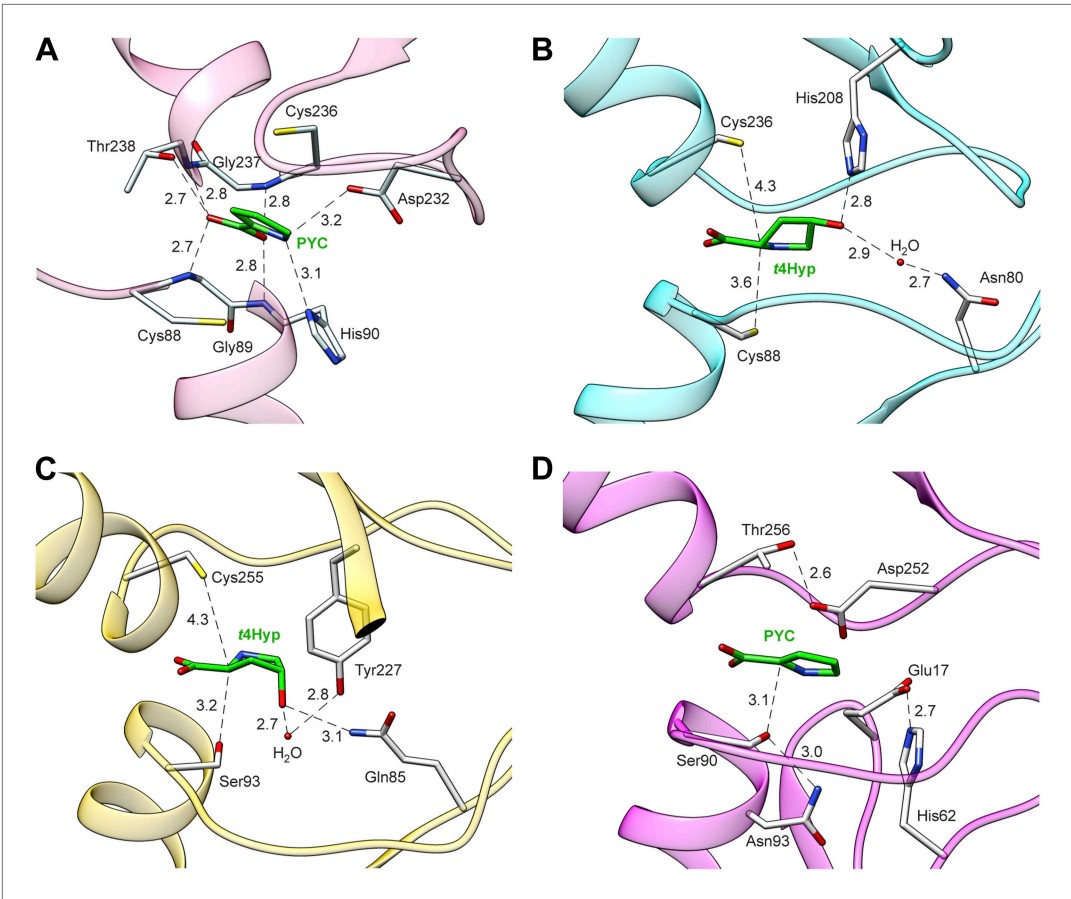

**Figure 5**. Structures of members of the PRS. (**A**) Structure of Q4KGU2 (locus tag: PFL_1412; cluster 2) with PYC illustrating the utilization of the carboxyl group to bridge the N-terminal amide backbone groups of two opposing α-helices. While In B9K4G4 (**D**) and B9JQV3 (**C**) the relative positions of residues that coordinate the prolyl nitrogen (Asp 232, His 90) are conserved His 90 is replaced by a Ser. (**B**) Structure of Q4KGU2 with t4Hyp illustrating the interactions Q4KGU2 with the 4-hydroxyl group and the relative positions of the two catalytic cysteine residues. (**C**) Structure of B9JQV3 (locus tag: Avi_0518, cluster 9) with t4Hyp illustrating the interactions of B9JQV3 with the 4-hydroxyl group of t4Hyp and the relative positions of the catalytic Ser (Ser 93, trans→cis) and Cys (Cys 236, cis→trans). (**D**) Structure of B9K4G4 (Avi_7022, cluster 3) with PYC illustrating the position of the catalytic Ser (Ser 90, dehydration), and the non-catalytic orientation of Thr 256 which replaces the Cys observed in Cys/Cys containing PRS members. In addition, the catalytic Ser (Ser 90) is positioned by hydrogen bonding interactions between the side chain of Asn 93 (shown) and the backbone nitrogen of Asn 93 (not shown). Based on this work, all ProR family members with a catalytic Ser at this position (including B9JQV3, determined here) are proposed to have this motif.

proline ketimine (*Table 7*). Thus, the GNN provided essential information for predicting/assigning functions to the members of the light sky blue cluster in the PRS SSN.

## Discussion

Although in most cases interpretations of the functional relationships of the clusters in the GNN with those in the query SSN are straightforward, complications can arise. For example, in several species, two members of the PRS are encoded by proximal genes, that is, a 4HypE and a t3HypD; these species can utilize both t4Hyp and *trans*-3-hydroxy-L-proline as carbon and nitrogen sources. Thus, the GNN contains a cluster for the PRS (right-hand cluster in the top row [when used as query, each PRS finds the adjacent PRS; *Figure 3I*]). For these species, clusters in the GNN are a composite of two genome contexts, that is, the proteins/enzymes that participate in both catabolic pathways. These situations can be deconvoluted by coloring the nodes identified by two queries with the colors for both query clusters in the GNN. With the genome contexts/metabolic pathways identified for 'genome-isolated' 4HypEs and t3HypDs, this complication is easy to identify and understand.

The GNN also is useful to assess the physiological importance of in vitro promiscuity. Several of the purified proteins catalyze both the 4HypE and t3HypD reactions (**Tables 1 and 2**). Some of these promiscuous proteins identify both the OCD or MLD2 superfamilies (predicting the t3HypD pathway) and the DAAO, DHDPS, and aldehyde dehydrogenase superfamilies (predicting the 4HypE pathway) in their genome neighborhoods (**Figure 7**). In these cases, we conclude that the in vitro promiscuity is not an 'artifact' but is physiologically significant.

As established in this study, the majority of the members of the PRS catalyze only the three previously characterized (known) reactions (**Figure 1**). As a result, we were able to use the GNN without any additional information to correctly predict functions for all of the highly populated clusters/families (>85% of the members; **Figure 8**). Because of this simplicity, the PRS provides a

**Table 6.** Data Collection and Refinement Statistics[a]

| UNIPROT / CLUSTER / PROTEIN | A5VZY6 / 2 / Pput_1285 | A5VZY6 / 2 / Pput_1285 | Q1GU06 / 2 / Csal_2705 | Q8P833 / 2 / XCC_2415 | B2D6W2 / 2 / BMULJ_04063 |
|---|---|---|---|---|---|
| Organism | Pseudomonas putida F1 | Pseudomonas putida F1 | Chromohalobacter salexigens DSM 3043 | Xanthomonas campestris | Burkholderia multivorans |
| PDBID | 4JBD | 4JD7 | 4JCI | 4JUU | 4K7X |
| **DIFFRACTION DATA STATISTICS** | | | | | |
| Space Group | I2 | $P2_12_12_1$ | $P2_12_12_1$ | $P2_12_12_1$ | $I4_122$ |
| Unit Cell (Å , °) | a=45.2 b=54.2 c=142.7 | a=64.8 b=96.8 c=109.2 | a=48.1 b=54.4 c=253.0 | a=54.9 b=108.8 c=116.2 | a=114.9 b=114.9 c=173.7 |
| Resolution (Å) | 1.3 (1.3-1.32) | 1.5 (1.5-1.58) | 1.7 (1.7-1.79) | 1.75 (1.75-1.84) | 1.75 (1.75-1.84) |
| Completeness (%) | 99.8 (99.6) | 99.5 (98.9) | 97.0 (94.0) | 99.7 (99.4) | 100.0 (100.0) |
| Redundancy | 3.6 (3.5) | 7.3 (7.1) | 9.3 (7.8) | 7.3 (7.1) | 14.3 (13.5) |
| Mean(I)/sd(I) | 7.9 (1.4) | 18.0 (1.1) | 17.5 (3.3) | 18.0 (1.1) | 14.1 (1.1) |
| $R_{sym}$ | 0.062 (0.735) | 0.067 (0.707) | 0.073 (0.644) | 0.074 (0.725) | 0.130 (0.699) |
| **REFINEMENT STATISTICS** | | | | | |
| Resolution (Å) | 1.3 (1.3-1.31) | 1.5 (1.5-1.52) | 1.7 (1.7-1.72) | 1.75 (1.75-1.77) | 1.75 (1.75-1.78) |
| Unique reflections | 82749 | 109888 | 72128 | 70700 | 58574 |
| $R_{cryst}$ (%) | 15.8 (30.4) | 15.9 (22.6) | 17.1 (23.7) | 15.2 (21.5) | 13.8 (19.7) |
| $R_{free}$ (%, 5% of data) | 18.4 (31.1) | 17.5 (25.4) | 20.5 (26.2) | 18.4 (26.4) | 15.6 (18.5) |
| Residues In Model [Expected] | A1-A308 [1-308] | A(-5)-A308, D(-3)-D308 [1-308] | A(-3)-A169, A171-A309 [1-311] | A(-2)-A312, B(-2)-B312 [1-312] | A(-3)-A310 [1-311] |
| Residues / Waters / Atoms total | 308 / 453/ 3142 | 626 / 752 / 6225 | 620 / 494 / 5780 | 626 / 596 / 5841 | 314 / 463 / 3223 |
| Bfactor Protein/Waters/Ligand | 17.3 / 31.2 / 21.7 | 19.3 / 30.5 / 27.9 | 24.8 / 33.6 / - | 23.9 / 35.2 / 37.3 | 15.6 / 34.0 / 30.6 |
| Ligand | Citrate | Sulfate | - | Phosphate / UNL | Phosphate |
| RMSD Bond Lengths (Å) / Angles (°) | 0.008 / 1.283 | 0.009 / 1.325 | 0.011 / 1.332 | 0.010 / 1.26 | 0.009 / 1.268 |
| Ramachandran Favored / Outliers (%) | 98.7 / 0.0 | 96.8 / 0.00 | 98.2 / 0.00 | 99.0 / 0.0 | 97.7 / 0.0 |
| Clashscore [b] | 2.32 (99th pctl) | 3.02 (98th pctl) | 3.74 (97th pctl) | 4.14 (97th pctl) | 3.12 (97th pctl) |
| Overall score [b] | 1.01 (99th pctl) | 1.29 (95th pctl) | 1.16 (99th pctl) | 1.22 (99th pctl) | 1.16 (99th pctl) |

[a] Data in parenthesis is for the highest resolution bin

[b] Scores are ranked according to structures of similar resolution as formulated in MOLPROBITY

*Table 6. Continued on next page*

*Table 6. Continued*

| UNIPROT / CLUSTER / PROTEIN | Q4KGU2 / 2 / PFL_1412 | Q4KGU2 / 2 / PFL_1412 | A6WW16 / 3 / Oant_0439 | B9K4G4 / 3 / Avi_7022 | B9JQV3 / 9 / Avi_0518 |
|---|---|---|---|---|---|
| Organism | Pseudomonas fluorescens Pf-5 | Pseudomonas fluorescens Pf-5 | Ochrobacterium anthropi | Agrobacterium vitis S4 | Agrobacterium vitis S4 |
| PDBID | 4J9W | 4J9X | 4K8L | 4K7G | 4LB0 |
| **DIFFRACTION DATA STATISTICS** | | | | | |
| Space Group | $P2_1$ | $P2_12_12_1$ | $I222$ | $P4_32_12$ | $P4_22_12$ |
| Unit Cell (Å , °) | $a=56.2$ $b=74.6$ $c=87.1$ $\beta=105.5$ | $a=64.8$ $b=96.8$ $c=109.2$ | $a=77.3$ $b=78.3$ $c=114.4$ | $a=54.9$ $b=108.8$ $c=116.2$ | $a=178.0$ $b=178.0$ $c=49.7$ |
| Resolution (Å) | 1.6 (.6-1.69) | 1.7 (1.7-1.79) | 1.9 (1.9-2.0) | 2.0 (2.0-2.1) | 1.7 (1.7-1.79) |
| Completeness (%) | 99.3 (99.5) | 99.5 (99.0) | 99.8 (100.0) | 100 (100) | 99.9 (99.9) |
| Redundancy | 3.6 (3.5) | 6.7 (6.0) | 7.2 (7.3) | 14.1 (13.2) | 10.4 (7.9) |
| Mean(I)/sd(I) | 6.9 (1.7) | 11.6 (1.5) | 6.0 (1.3) | 11.6 (3.3) | 18.3 (2.7) |
| $R_{sym}$ | 0.093 (0.434) | 0.088 (0.531) | 0.09 (0.594) | 0.17 (0.836) | 0.078 (0.745) |
| **REFINEMENT STATISTICS** | | | | | |
| Resolution (Å) | 1.6 (1.6-1.62) | 1.7 (1.7-1.72) | 1.9 (1.9-1.97) | 2.0 (2.0-2.02) | 1.7 (1.72-1.70) |
| Unique reflections | 90740 | 77405 | 27674 | 86628 | 87548 |
| $R_{cryst}$ (%) | 19.7 (28.8) | 19.4 (23.5) | 16.8 (17.6) | 13.6 (19.5) | 15.8 (22.9) |
| $R_{free}$ (%, 5% of data) | 23.2 (33.8) | 22.5 (27.5) | 20.7 (21.7) | 16.6 (22.9) | 19.2 (27.3) |
| Residues In Model [Expected] | A1-A310, B1-B310 [1-310] | A1-310, B1-310 [1-310] | A0-A157, A161-A184, A193-A245, A255-280, A289-A332 [1-343] | B5-B342, D(-9)-D342 [1-342] | A1-A323, A326-A344, B0-B346 [1-347] |
| Residues / Waters / Atoms total | 620 / 537 / 5301 | 620 / 630 / 5378 | 305 / 191 / 2824 | 690 / 780 / 6761 | 689 / 701 / 6633 |
| Bfactor Protein/Waters/Ligand | 21.1 / 32.2 / 12.9 | 22.9 / 34.0 / 16.3 | 31.3 / 37.7 / - | 24.1 / 37.5 / 15.2 | 25.1 / 36.2 / 17.9 |
| Ligand | (PYC) Pyrrole 2-carboxylate | (t4Hyp) Trans-4OH-L-Proline | - | (PYC) Pyrrole 2-carboxylate | (t4Hyp) Trans-4OH-L-Proline / Acetate |
| RMSD Bond Lengths (Å) / Angles (°) | 0.006 / 1.079 | 0.006 / 1.093 | 0.011 / 1.349 | 0.011 / 1.311 | 0.010 / 1.320 |
| Ramachandran Favored / Outliers (%) | 98.7 / 0.0 | 98.5 / 0.0 | 98.3 / 0.0 | 98.0 / 0.3 | 98.4 / 0.3 |
| Clashscore [b] | 1.59 (99th pctl) | 1.82 (99th pctl) | 6.6 (93rd pctl) | 2.8 (99th pctl) | 2.2 (99th pctl) |
| Overall score[b] | 0.97 (100th pctl) | 0.94 (100th pctl) | 1.36 (98th pctl) | 1.08 (100th pctl) | 1.0 (100th pctl) |

[a] Data in parenthesis is for the highest resolution bin
[b] Scores are ranked according to structures of similar resolution as formulated in MOLPROBITY

lucid illustration of the strategy by which a query SSN and its GNN can be used to predict and assign enzymatic functions.

However, large-scale prediction and assignment of function to members of many functionally diverse (super)families will be more complicated than that described for the PRS and require information from complementary experimental and computational approaches. The use of GNNs is restricted to those enzymes that are encoded by proximal operons and/or gene clusters in eubacteria and archaea. For *Escherichia coli* K-12, 60% of the genes are located in polycistronic transcriptional units that may provide linked functional information that can be used to identify pathways; 40% are located in monocistronic transcriptional units (http://regulondb.ccg.unam.mx/menu/tools/regulondb_overviews/chart_form.jsp).

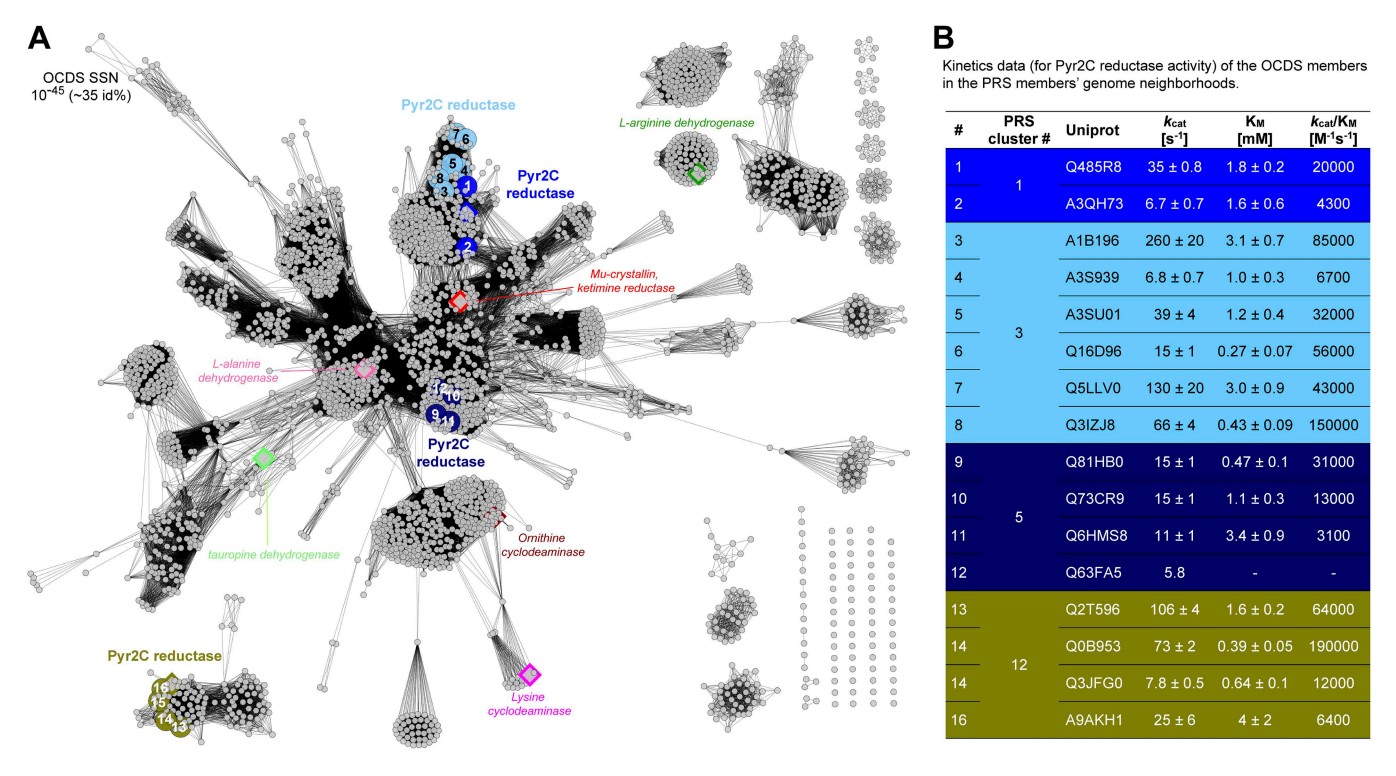

**Figure 6.** Sequence divergent members of the ornithine cyclodeaminase superfamily (OCDS) have been the assigned novel pyrroline-2-carboxylate reductase (Pyr2C reductase) function in this work. (**A**) The OCDS SSN displayed at the e-value cutoff $10^{-45}$ (~35% sequence identity). The Pyr2C reductase function is located in four clusters; these proteins are shown in large colored circles, labeled from 1 to 16, and color-coded by the colors of the PRS query sequences shown in *Figure 2B*. Proteins representing several previously characterized functions in the OCDS are shown by large diamonds, with borders in hotpink (L-alanine dehydrogenase [*Schröder et al., 2004*]), brown (ornithine cyclodeaminase [*Goodman et al., 2004*]), magenta (lysine cyclodeaminase [*Gatto et al., 2006*]), red (ketamine reductase [*Hallen et al., 2011*]), green (L-arginine dehydrogenase [*Li and Lu, 2009*]) and palegreen (tauropine dehydrogenase [*Kan-No et al., 2005*; *Plese et al., 2008*]), respectively. Their annotations are shown in italics. The diamonds with blue and olive borders are Pyr2C reductases recently characterized by *Watanabe et al. (2014)*. (**B**) Kinetics data for the Pyr2C reductase activity for the 16 members of the OCDS shown in panel **A** using NADPH as the cosubstrate.

Thus, genome neighborhood context is not a general solution to infer functions for many proteins/enzymes of unknown function encoded eubacterial and archaeal genomes. Even for those proteins encoded by polycistronic transcriptional units, complete metabolic pathways may be encoded by multiple transcriptional units (mono- and/or polycistronic) that are not genome proximal; these pathways and their component enzymes and ligand binding proteins (solute binding proteins for transport systems and transcriptional regulators) may be recognized by regulon analyses that identify conserved binding sites for transcriptional regulators (*Ravcheev et al., 2013*; *Rodionov et al., 2013*).

To the extent that genome neighborhoods and/or regulons allow the identification of the components of unknown/novel metabolic pathways, the locations of these proteins/enzymes in the SSNs for their (super)families will provide restrictions on their ligand/substrate specificities and/or reaction mechanisms (*Atkinson et al., 2009*). Also, as we recently demonstrated (*Zhao et al., 2013*), in silico (virtual) docking of ligand libraries to multiple binding proteins and enzymes in an unknown metabolic pathway (pathway docking) is a powerful approach to enhance the reliability of docking to predict novel ligand/substrate specificities and identify novel metabolic pathways

Irrespective of the many complications associated with assignment of function to unknown proteins/enzymes, we conclude that GNNs provide a novel approach for large-scale analysis and visualization of genome neighborhood context in enzyme (super)families. We are continuing to improve the use of GNNs as well as regulon analyses and pathway docking to facilitate the discovery of novel enzymes and the metabolic pathways in which they function.

**Table 7.** Kinetic constants for the proline ketimine reductases (members of the malate/Llactate dehydrogenase 2 [MLD2] and ornithine cyclodeaminase [OCD] superfamilies) that are in the genome neighborhoods of members of the PRS

| Cluster | UniProt | Locus tag | Cofactor | $k_{cat}$ [s$^{-1}$] | $K_M$ [mM] | $k_{cat}/K_M$[M$^{-1}$s$^{-1}$] |
|---|---|---|---|---|---|---|
| MLD2_PRS_light skyblue (3) | Q7CVK1 | Atu4676 | NADPH | 32 ± 1 | 0.33 ± 0.04 | 99000 |
| | Q9I492 | PA1252 | NADPH | 1.6 ± 0.05 | 0.41 ± 0.06 | 3900 |
| MLD2_PRS_Red (2) | Q4KGT8 | PFL_1416 | NADPH | 20 ± 0.8 | 1.1 ± 0.2 | 18000 |
| | Q0B9S2 | Bamb_3547 | NADPH | 54 ± 13 | 9.4 ± 4 | 5700 |
| | A9ALD3 | Bmul_4451 | NADPH | 33 ± 2 | 7.4 ± 1 | 4400 |
| MLD2_PRS_indigo (13) | Q4KAT3 | PFL_3547[a] | NADPH | - | - | 2300[b] |
| OCD_PRS_light skyblue (3) | A1B196 | Pden_1185 | NADPH | 260 ± 20 | 3.1 ± 0.7 | 85000 |
| | | | NADH | 81 ± 20 | 16 ± 6 | 5100 |
| | A3S939 | EE36_06353[a] | NADPH | 6.8 ± 0.7 | 1.0 ± 0.3 | 6700 |
| | A3SU01 | NAS141_11281[a] | NADPH | 39 ± 4 | 1.2 ± 0.4 | 32000 |
| | | | NADH | 8.2 ± 4 | 73 ± 50 | 110 |
| | Q16D96 | RD1_0323[a] | NADPH | 15 ± 1 | 0.27 ± 0.07 | 56000 |
| | | | NADH | 3.7 ± 0.4 | 11 ± 3 | 320 |
| | Q5LLV0 | SPO3821[a] | NADPH | 130 ± 20 | 3.0 ± 0.9 | 43000 |
| | | | NADH | - | - | 840[b] |
| | Q3IZJ8 | RSP_0854[a] | NADPH | 66 ± 4 | 0.43 ± 0.09 | 150000 |
| | | | NADH | 12[c] | - | - |
| OCD_PRS_navy (5) | Q81HB0 | BC_0906 | NADPH | 15 ± 1 | 0.47 ± 0.1 | 31000 |
| | | | NADH | 19 ± 1 | 11 ± 2 | 1800 |
| | Q73CR9 | BCE_0995 | NADPH | 15 ± 1 | 1.1 ± 0.3 | 13000 |
| | | | NADH | 2.1 ± 0.3 | 7.6 ± 3 | 270 |
| | Q6HMS8 | BT9727_0800 | NADPH | 11 ± 1 | 3.4 ± 0.9 | 3100 |
| | | | NADH | 2.1 ± 0.4 | 18 ± 6 | 120 |
| | Q63FA5 | BCE33L0803 | NADPH | 5.8[c] | - | - |
| | | | NADH | 0.87 ± 0.1 | 4.9 ± 2 | 180 |
| OCD_PRS_olive (12) | Q0B953 | Bamb_3766 | NADPH | 106 ± 4 | 1.6 ± 0.2 | 64000 |
| | | | NADH | 41 ± 6 | 7.3 ± 3 | 5700 |
| | Q2T596 | BTH_II1457[a] | NADPH | 73 ± 2 | 0.39 ± 0.05 | 190000 |
| | | | NADH | 203 ± 23 | 32 ± 7 | 6400 |
| | Q3JFG0 | BURPS1710b_A2543[a] | NADPH | 7.8 ± 0.5 | 0.64 ± 0.1 | 12000 |
| | | | NADH | 6.0 ± 1 | 31 ± 13 | 190 |
| | A9AKH1 | Bmul_4263 | NADPH | 25 ± 6 | 4 ± 2 | 6400 |
| OCD _PRS_blue (1) | Q485R8 | CPS_1455 | NADPH | 35 ± 0.8 | 1.8 ± 0.2 | 20000 |
| | | | NADH | - | - | 170[b] |
| | A3QH73 | Shew_2955[a] | NADPH | 6.7 ± 0.7 | 1.6 ± 0.6 | 4300 |
| | | | NADH | 0.37 ± 0.1 | 26 ± 10 | 14 |

[a]Highly homologous to MLD2 or OCD which are in the gene context of proline racemase. [b] The enzyme didn't saturate. [c] $K_M$ is too small (< 0.03mM).

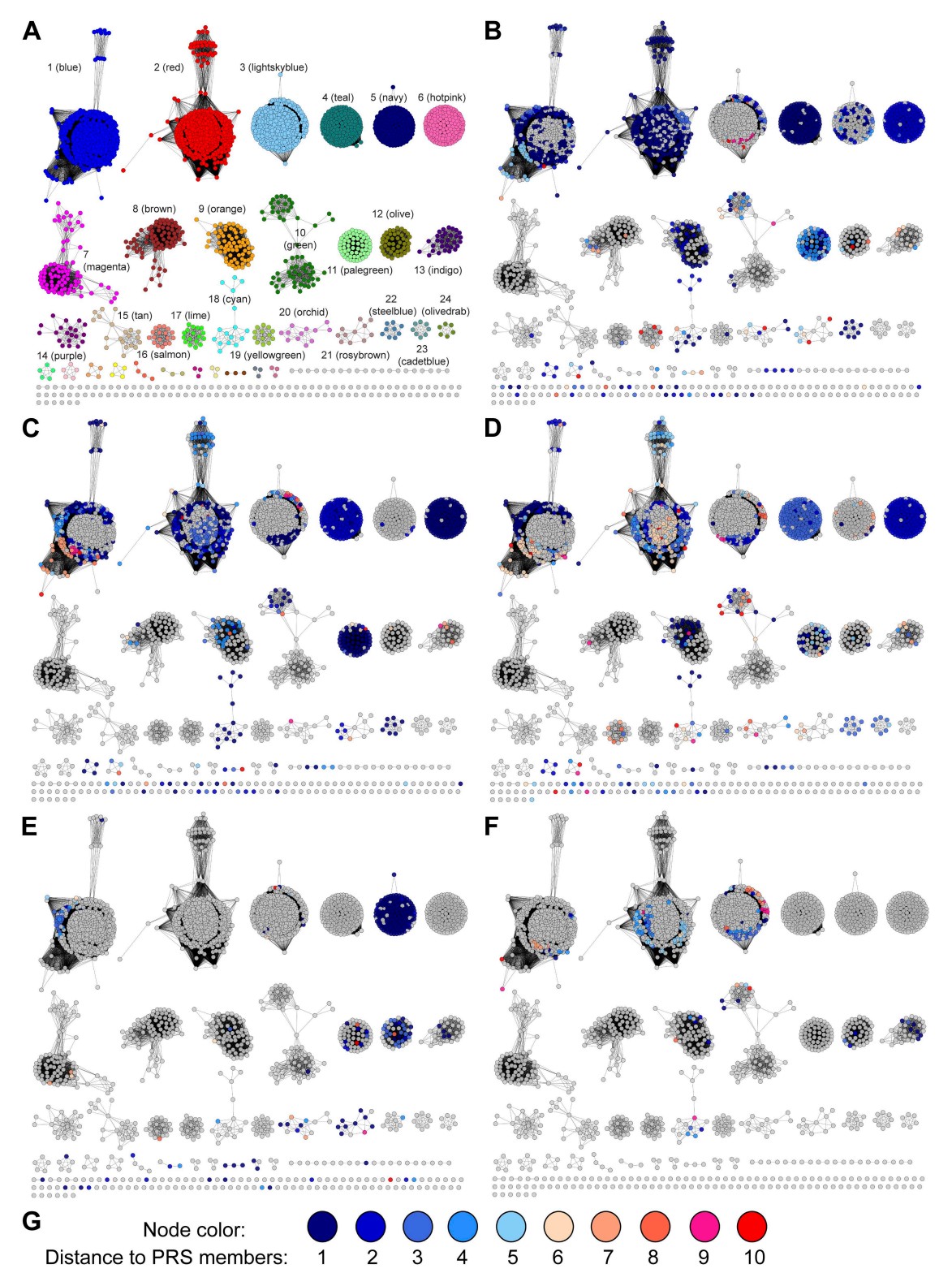

**Figure 7**. Mapping members of GNN clusters back to the SSN for the PRS. (**A**) SSN for the PRS with cluster numbers. (**B**) D-amino acid oxidase (DAAO). (**C**) Dihydrodipicolinate synthase (DHDPS). (**D**) Aldehyde dehydrogenase. (**E**) Ornithine cyclodeaminase (OCD). (**F**) Malate/L-lactate dehydrogenase 2 (MLD2). (**G**) The color scheme for **B–F**.

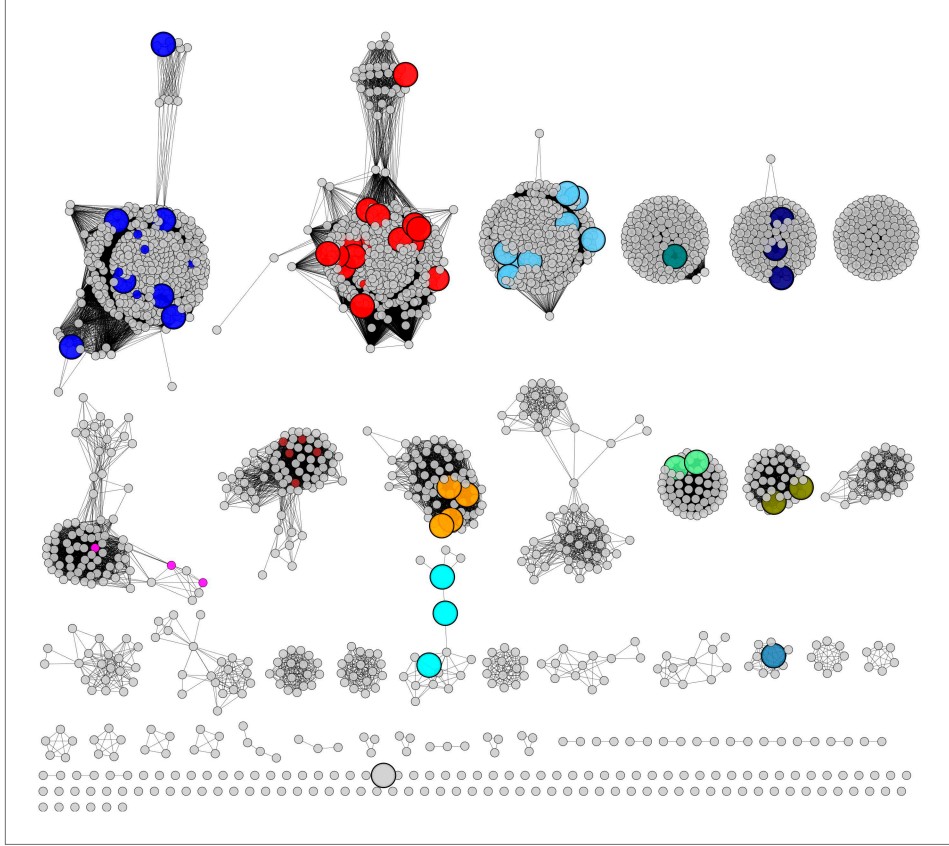

**Figure 8**. Experimentally characterized enzymes reported by Swiss-Prot (small colored circles) and newly characterized in this work (large colored circles). Colors match the color scheme in *Figure 2B*.

## Materials and methods

### Sequence similarity networks (SSN)

The SSNs for the PRS (*Figure 2*) and the OCDS (*Figure 5A*) were created using Pythoscape v1.0 (*Barber and Babbit, 2012*) that is available for download from http://www.rbvi.ucsf.edu/trac/Pythoscape The input sequences were downloaded from the InterPro webpages of PRS and OCDS: http://www.ebi.ac.uk/interpro/entry/IPR008794, http://www.ebi.ac.uk/interpro/entry/IPR003462, respectively. Cytoscape v2.8 (*Smoot et al., 2011*) is used for visualization and analysis of the SSN.

### Genome neighborhood network (GNN)

The GNN for the PRS (*Figure 3*) was also created using Pythoscape v1.0 (*Barber and Babbit, 2012*). At an e-value cutoff $10^{-110}$, each cluster in the SSN was assigned a unique cluster number and color, which are used for labeling and coloring genome context sequences. Genome context sequences were collected from the ±10 gene range of each PRS member and used as the input sequences for making the GNN using the procedure for generating a SSN.

### Protein production

Genes for members of the PRS that are encoded by the genomic DNAs in the Macromolecular Therapeutics Development Facility at the Albert Einstein College of Medicine were cloned into pNIC28-BSA4-based expression vectors as previously described (*Sauder et al., 2008*).

### Protein expression

The pNIC28-BSA4-based expression plasmids were transformed into *Escherichia coli* BL21(DE3) containing the pRIL plasmid (Stratagene, Agilent Technologies, Inc., Wilmington, DE) and used to inoculate 20 ml 2xYT cultures containing 50 µg/ml kanamycin and 34 µg/ml chloramphenicol. Cultures were

allowed to grow overnight at 37°C in a shaking incubator; these were used to inoculate 2 L of PASM-5052 auto-induction medium (Studier). The cultures were placed in a LEX48 airlift fermenter and incubated at 37°C for 5 hr and then at 22°C overnight (16–20 hr). The cells were collected by centrifugation at 6000×*g* for 10 min and stored at −80°C.

## Purification of proteins

Cells were resuspended in Lysis Buffer (20 mM HEPES, pH 7.5, containing 20 mM imidazole, 500 mM NaCl, and 5% glycerol) and lysed by sonication. Lysates were clarified by centrifugation at 35,000×*g* for 45 min. The clarified lysates were loaded on a 1-ml His60 Ni-NTA column (Clontech) using an AKTAxpress FPLC (GE Healthcare). The columns were washed with 10 column volumes of Lysis Buffer and eluted with buffer containing 20 mM HEPES, pH 7.5, containing 500 mM NaCl, 500 mM imidazole, and 5% glycerol. The purified proteins were loaded onto a HiLoad S200 16/60 PR gel filtration column equilibrated with a buffer containing 20 mM HEPES, pH 7.5, 150 mM NaCl, 5% glycerol, and 5 mM DTT. The purities of the proteins were analyzed by SDS-PAGE. The proteins were snap frozen in liquid $N_2$ and stored at −80°C.

## Crystallization

Proteins were screened for crystallization conditions using commercially available screens (MCSG 1, 2, and 4 [Microlytic, Woburn MA] and MIDAS [Molecular Dimensions, Altamonte Springs FL]) using sitting drop vapor diffusion 96-well INTELLIPLATES (Art Robbins Instruments, Sunnyvale CA), a PHOENIX crystallization robot (Art Robbins Instruments), and stored and monitored in a Rock Imager 1000 (Formulatrix, Waltham MA) plate hotel. Protein (1 µl) was combined with an equivalent volume of precipitant and equilibrated against a 70 µl reservoir of the same precipitant at room temperature (~292 K).

**A5VZY6**, (27.9 mg/mL, 15 mM HEPES, pH 7.5, containing 150 mM NaCl, and 5 mM DTT) was crystallized in 0.1 M sodium acetate, pH 4.6, containing 1.5 M $LiSO_4$; the crystals grew as rectangular bricks over a 1-week period (SPG-P2₁2₁2₁). For the cryoprotectant, the $LiSO_4$ concentration was increased to 1.8M.

**A5VZY6** was also crystallized (27.9 mg/ml, 15 mM HEPES, pH 7.5, containing 150 mM NaCl, and 5 mM DTT) in 0.2 M diammonium hydrogen citrate pH 5.0, containing 20% (wt/vol) PEG 3350; the crystals grew as wedges over a 1-week period. The cryoprotectant contained 20% glycerol.

**Q1QU06** (21.1 mg/ml, 15 mM HEPES, pH 7.5, containing 150 mM NaCl, and 5 mM DTT) was crystallized in 0.2 M di-ammonium hydrogen citrate, pH 5.0, containing 20% (wt/vol) PEG 3350; the crystals grew as plates over 2–3 days. The cryoprotectant contained 20% glycerol.

**XCC2415** (29.3 mg/ml, 15 mM HEPES, pH 7.5, containing 150 mM NaCl, and 5 mM DTT) was crystallized in 0.1 M HEPES, pH 7.5, containing 0.8 M sodium phosphate and 0.8 M potassium phosphate and grew as thin rods over 2–3 days. The cryoprotectant contained 20% glycerol.

**B3D6W2** (21.8 mg/ml, 15 mM HEPES, pH 7.5, containing 150 mM NaCl, and 5 mM DTT) was crystallized in 0.1 M phosphate-citrate, pH 4.2, containing 1.6 M $NaH_2PO_4$, and 0.4 M $K_2HPO_4$ and grew as large rods over 2 weeks. The cryoprotectant contained 20% glycerol.

**Q4KGU2** (25.7 mg/ml, 15 mM HEPES, pH 7.5, containing 150 mM NaCl, and 5 mM DTT) was crystallized in 0.2 M ammonium acetate, 0.1 M trisodium citrate, pH 5.6, containing 14% PEG4000, 5% glycerol, and either 20 mM PYC or 50 mM *t*4Hyp and grew as thick plates over 2–3 days. The cryoprotectant contained 20% glycerol.

For **A6WW16**, **B9K4G4**, and **B9JQV3**, TEV protease (*Tropea et al., 2009*) was added at a 1/80 ratio prior to crystallization setup. The samples were incubated on ice for 2 hr, and the buffer was exchanged with 15 mM HEPES, pH 7.5, containing 5 mM DTT by dilution and centrifugal filtration. The extent of TEV cleavage was not measured.

**A6WW16** (17.3 mg/ml, 15 mM HEPES, pH 7.5, containing 5 mM DTT) was crystallized in 0.2 M sodium nitrate and 20% PEG3350 and grew as leaf petals over 2 to 3 weeks. The cryoprotectant contained 20% glycerol.

**B9K4G4**, (17.1 mg/ml, 15 mM HEPES, pH 7.5, containing 5 mM DTT) was crystallized in 0.1 M sodium acetate, pH 4.6, containing 1 M ammonium citrate and 25 mM pyrrole 2-carboxylate. Crystals grew from an initial precipitate as multifaceted crystals over a month. The cryoprotectant contained 20% glycerol.

**B9JQV3** (30.0 mg/ml, 15 mM HEPES, pH 7.5, containing 5 mM DTT) was crystallized in 0.1 M sodium acetate, containing 25% Peg4000, 8% 2-propanol, and 200 mM *t*4Hyp and grew as tetragonal rods over 2–3 days. The cryoprotectant contained 20% 2-propanol.

## Structure determination

Diffraction data were collected on beamline 31-ID (LRL-CAT, Advanced Photon Source, Argonne National Laboratory, IL) from single crystals at 100 K and a wavelength of 0.9793 Å. Data were integrated using MOSFLM (*Battye et al., 2011*) and scaled in SCALA (*Evans, 2006*).

Suitable molecular replacement models existed for all of the protein targets of this study. These included, 2AZP, a putative 4HypE (from cluster 2) determined unliganded by the Midwest Center for Structural Genomics, and 1TM0 (*Forouhar et al., 2007*), a putative *t*3HypD (cluster 3, also similar to cluster 9) with an unliganded and disordered active site, determined by the Northeast Structural Genomics Consortium. Molecular replacement computations were performed in AMORE (*Navaza, 1994*) utilizing the structure that exhibited the greatest homology to the target. If this was unsuccessful, either due to the particular issues with the space group, asymmetric unit composition, or a different orientation of the two domains, molecular replacement was performed with each of the domains separately within PHENIX (*Adams et al., 2004*; *Zwart et al., 2008*).

Iterative cycles of manual rebuilding within COOT (*Emsley and Cowtan, 2004*) and refinement within PHENIX were performed until the entire sequence was modeled. Inclusion of ligands, TLS (translation/libration/screw) refinement (domains chosen automatically within PHENIX) (*Winn et al., 2001*; *Painter and Merritt, 2006*) and editing of the solvent structure were performed in the final refinement cycles.

With one exception, the entire sequences of all of the targets could be modeled, except for a small number of residues at the N- or C-termini. The one outlier was **A6WW16** that had several disordered regions around the active site similar to the previously determined structure from this cluster (1TM0, cluster 3, light sky blue). Due to the relatively weak binding of the proline racemase family members for their substrates, inhibitors and substrates were included at high concentrations (25–200 mM). Even at these concentrations, several structures were determined from cluster 2 that bound anionic ligands (phosphate, citrate, etc) from the crystallization medium rather than the co-crystallized ligand, and the degree of domain closure about that ligand varied. For all of the structures liganded with either PYC or *t*4Hyp, the structures are determined in a closed state with Cα–Cα distances of 7–8 Å for the opposing active site catalytic Cys–Cys (cluster 2, red), Ser–Thr (cluster 3, light sky blue) or Ser–Cys dyad (cluster 9, orange). In the case of **Q4KGU2**, the ligand was *t*4Hyp state based on the electron density. In contrast, for **B9JQV3**, the density for the ligand had significant planer character, suggesting a mixture of *t*4Hyp and *c*4Hyp.

## ESI-MS screening of ProR, 4HypE, and t3HypD activities

Enzyme activity was screened by the mass change resulting from racemization /epimerization (+1 peak shift) and/or dehydration (−17 peak shift) for reactions in $D_2O$. Each enzyme (1 μM) was incubated with substrate libraries (*Table 1*) containing proline and proline betaine derivatives (0.1 mM each) along with 20 mM ammonium bicarbonate in $D_2O$ at a final volume of 200 μl at 30°C for 16 hr. 50 μl of the reaction mixture was aliquoted and dried with an Eppendorf vacufuge concentrator. The residue was suspended in 10 μl of $H_2O$, and 5 μl of the solution was mixed with the 5 μl of 50% methanol containing 0.4% (vol/vol) formic acid. A 10 μl sample was analyzed for ESI-MS.

## $^1$H NMR assay to confirm PRS reactions

If a change in mass was observed in the ESI-MS screening assays, a $^1$H NMR assay was performed to determine the product. Each reaction mixture contained 1 μM enzyme, 10 mM substrate, and 25 mM sodium phosphate buffer, pD 8, in a total volume of 800 μl $D_2O$. The mixture was incubated at 30°C for 16 hr before acquisition of the 500 MHz (*Hunter et al., 2012*) H NMR spectrum (*Figure 9*).

## Polarimetric assay to determine PRS kinetics

The enzyme activity was measured in a Jasco P-1010 polarimeter with a Hg 405-nm filter at 25°C by quantitating the change in optical rotation. The assay mixture contained 1 mM dithiothreitol (DTT) and 50 mM $Na^+$-phosphate buffer, pH 8.0.

## UV spectrophotometric assay for $\Delta^1$-Pyr2C reductase activity

$\Delta^1$-Pyr2C reductase assays were performed by measuring the decrease in the absorbance of NAD(P)H at 340 nm at 25°C with a Cary 300 Bio UV-Visible spectrophotometer (Varian). The reaction mixture (300 μl) contained variable concentrations of Pyr2C, 50 mM Tris–HCl buffer, pH 7.6, 0.16 mM NAD(P) H, and enzyme.

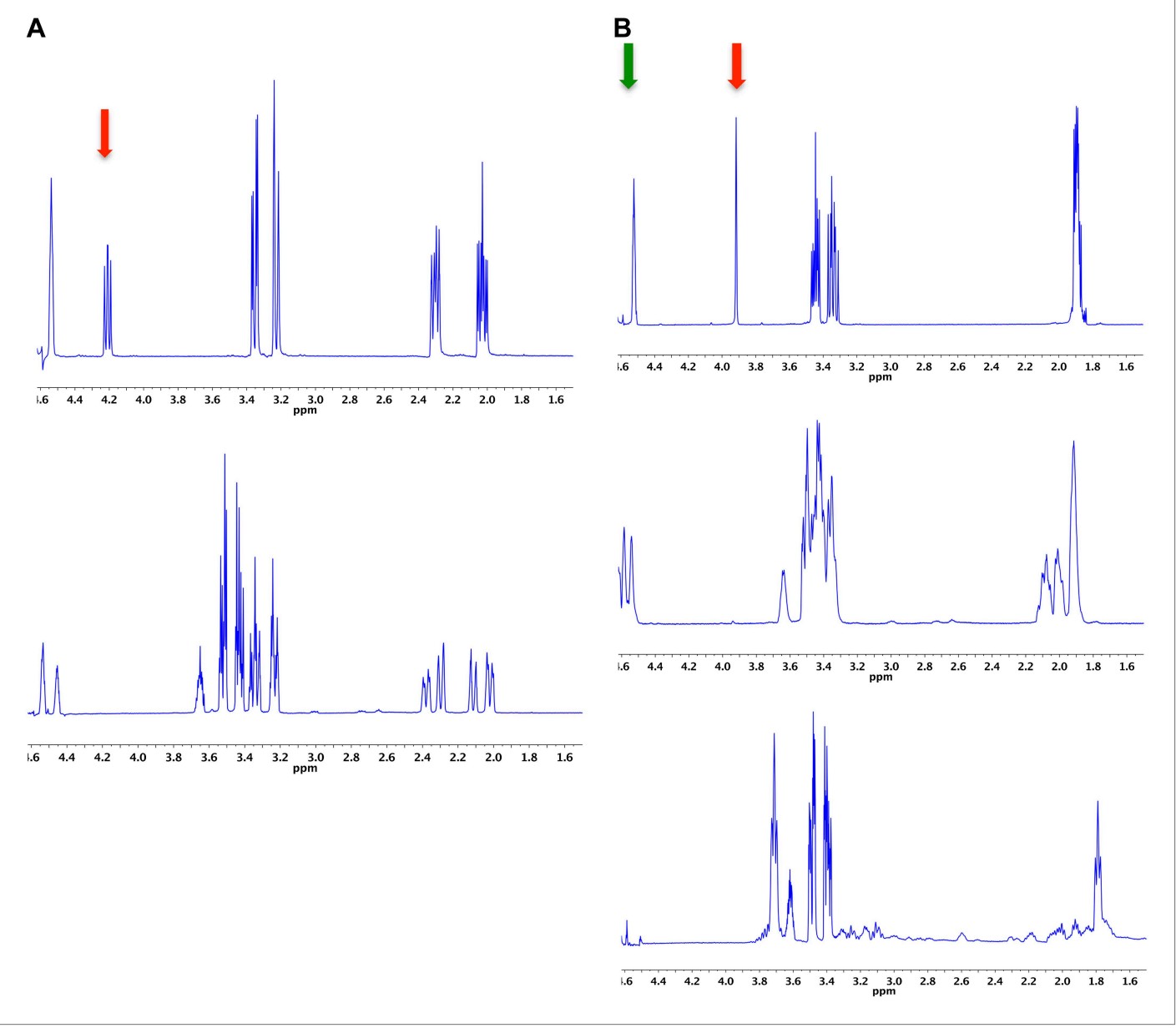

**Figure 9**. Demonstration of the 4HypE, 3HypE, and t3HypD reactions by $^1$H NMR. (**A**) $^1$H NMR spectra of the 4Hyp substrate mixture in 25 mM Na$^+$-phosphate buffer, pD 8, in D$_2$O (top) and 4Hyp mixture with **A3QFI1** (cluster 1, blue) showing 4Hyp epimerization (bottom). The red arrow indicates the proton at C2 for epimerization. The enzyme was stored in glycerol, so the spectra show resonances for glycerol between 3.4 and 3.7 ppm. (**B**) $^1$H NMR spectra of the t3Hyp substrate mixture in 25 mM Na$^+$-phosphate buffer, pD 8, in D$_2$O (top), t3Hyp mixture with D0B556 (cluster 3, light sky blue) showing 3Hyp epimerization (middle), and t3Hyp mixture with B9K4G4 (cluster 3, light sky blue) showing t3Hyp dehydration (bottom). The red arrow indicates the proton at C2 for epimerization; the green arrow indicates the proton at C3 for dehydration.

## $^1$H NMR assay for $\Delta^1$-Pyr2C reductase activity

The reaction mixture contained 10 mM $\Delta^1$-Pyr2C, 1 µM enzyme, 0.16 mM NADPH, 25 mM phosphate-Na buffer, pD 8.0, 1 U/ml alcohol dehydrogenase (NADP$^+$-dependent from *Thermoanaerobium brockii*, Sigma) and 80 µl isopropanol in a total volume of 800 µl of D$_2$O; the reaction was incubated at 30°C for 16 hr. The solvent was removed by lyophilization, 800 µl of D$_2$O was added, and the $^1$H NMR spectrum was recorded. Representative spectra are shown in *Figure 10*.

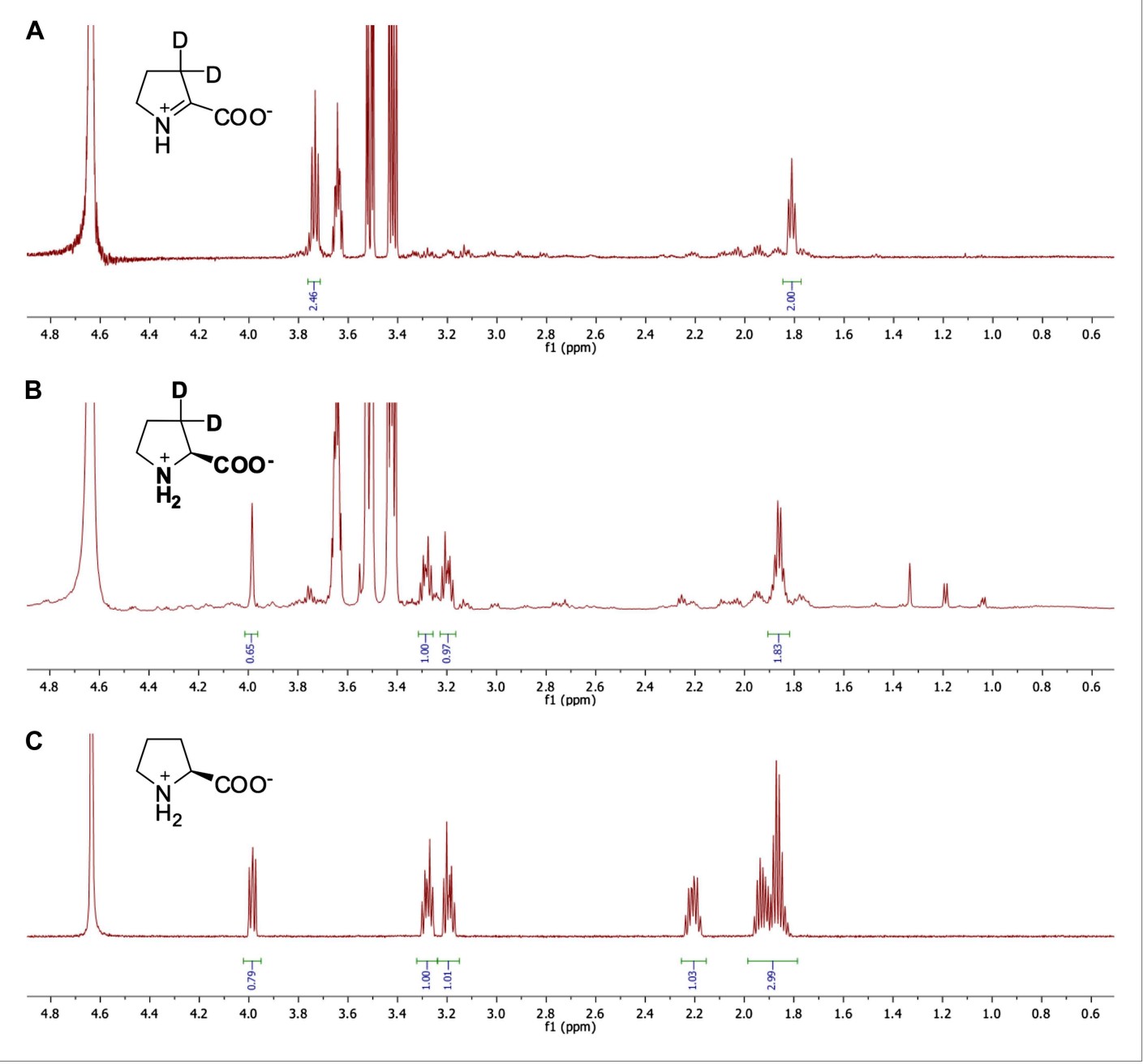

**Figure 10**. Representative $^1$H NMR spectra for $^{\triangle 1}$-pyrroline-2-carboxylate ($^{\triangle 1}$-Pyr2C) reductase activity. (**A**) $^1$H NMR spectrum of $^{\triangle 1}$-Pyr2C substrate in sodium phosphate, pD 8.0, in $D_2O$. (**B**) $^1$H NMR spectrum of Q7CVK1 (locus tag: Atu4676) incubated with $^{\triangle 1}$-Pyr2C, NADPH, and the cofactor regeneration system of alcohol dehydrogenase (NADP$^+$-dependent) and isopropanol in sodium phosphate, pD 8.0 in $D_2O$. (**C**) $^1$H NMR spectrum of L-proline in 25 mM sodium phosphate, pD 8.0, in $D_2O$.

## Bacterial strains and growth conditions

Bacterial strains are listed in *Table 8*. All strains were grown at 30°C with shaking at 225 rpm and were routinely cultured in Tryptic Soy Broth (Difco), supplemented with 30 g L$^{-1}$ sea salts (Sigma-Aldrich) for *Labrenzia aggregata* IAM12614 and *Roseovarius nubinhibens* ISM.

For gene expression analyses and carbon utilization studies, strains were cultured in the following defined media:

**Table 8.** Strains used in this study

| Organism |
| --- |
| *Agrobacterium tumefaciens* C58 |
| *Sinorhizobium meliloti* 1021 |
| *Labrenzia aggregata* IAM12614 |
| *Pseudomonas aeruginosa* PAO1 |
| *Paracoccus denitrificans* PD1222 |
| *Rhodobacter sphaeroides* 2.4.1 |
| *Rhodobacter sphaeroides* 2.4.1ΔRSP3519 |
| *Bacillus cereus* ATCC14579 |
| *Roseovarius nubinhibens* ISM |
| *Escherichia coli* MG1655 |
| *Streptomyces lividans* TK24 |

**Table 9.** Oligonucleotide primers used for construction of the RS3519 knock-out in *Rhodobacter sphaeroides* 2.4.1

| Oligo | Sequence (5′–3′) |
| --- | --- |
| RS3519F.KO | CATATGATGCGCGTTCAGGACGTGTATAACG |
| RS3519R.KO | GCTGAGCTCAGAGGACGAGGAAGCCCGCGTCC |

**Table 10.** qRT-PCR primers for transcriptional analysis of individual proline racemase superfamily members

| Oligo | Sequence (5′–3′) |
| --- | --- |
| Atu16s-F | GACACGGCCCAAACTCCTAC |
| Atu16s-R | GGGCTTCTTCTCCGACTACC |
| Atu0398-F | TCACCATTGAGAAGGCCAAT |
| Atu0398-R | GGTTGACGAGGTCCTTCAGA |
| Atu3953-F | CAGCTTCAGTGGCATCAGG |
| Atu3953-R | GTGTTGTGCCCAATGATCC |
| Atu4684-F | GAAGAGGCGCATGAGATTG |
| Atu4684-R | CGAAACCCAAAGCCTTGTT |
| Bc16s-F | CTCGTGTCGTGAGATGTTGG |
| Bc16s-R | TGTGTAGCCCAGGTCATAAGG |
| Bc0905-F | CTTCGCTGACGGACAAGTAGA |
| Bc0905-R | TGTACCGCTGTTACGGACAA |
| Bc2835-F | AACAGACCCGTGTCATCCTG |
| Bc2835-R | ACTAAGCCAGCCGGTGTATCT |
| La16s-F | TGGTGGGGTAAAGGCCTAC |
| La16s-R | TGGCTGATCATCCTCTCAGAC |
| La28492-F | TGTTGAAGACGAGGCCAAG |
| La28492-R | AAAAGCCGAGCTGTTCGTT |
| La28502-F | CGCGTAATCGACAGCCATA |

*Table 10. Continued on next page*

*Agrobacterium tumefaciens* C58 was cultured in M9 minimal medium (per liter: 12.8 g $Na_2HPO_4.7H_2O$, 3.0 g $KH_2PO_4$, 0.5 g NaCl, 1.0 g $NH_4Cl$); *B. cereus* ATCC 14579 was cultured in a modified Spizizen's minimal medium (*Spizizen., 1958*) (per liter: 2.0 g $(NH_4)_2SO_4$, 11.0 g $K_2HPO_4$, 6.0 g $KH_2PO_4$, 1.0 g sodium citrate.$2H_2O$).

*Streptomyces lividans* TK24 was cultured in a modified minimal medium of Hopwood (*Hopwood., 1967*) (per liter: 1.0 g $(NH_4)_2SO_4$, 0.5 g $K_2HPO_4$, 0.005 g $FeSO_4.7H_2O$). M9 minimal medium, and Spizizen's minimal medium were supplemented with the following trace metals (per liter: 0.003 mg $CuSO_4.5H_2O$, 0.025 mg $H_3BO_3$, 0.007 mg $CoCl_2.6H_2O$, 0.016 mg $MnCl_2.4H_2O$, 0.003 mg $ZnSO_4.7H_2O$, 0.3 mg $FeSO_4.7H_2O$). The minimal medium of Hopwood was supplemented with the following trace metals (per liter: 0.08 mg $ZnCl_2$, 0.4 mg $FeCl_3.6H_2O$, 0.02 mg $CuCl_2.2H_2O$, 0.02 mg $MnCl_2.4H_2O$, 0.02 mg $Na_2B_4O_7.10H_2O$, 0.02 mg $(NH_4)_6Mo_7O_{24}.4H_2O$).

All other strains were grown in the following defined medium (per liter: 17.0 g $K_2HPO_4$, 2.5 g $(NH_4)_2SO_4$, 2.0 g NaCl) supplemented with the following trace metals (0.3 mg $FeSO_4.7H_2O$, 0.003 mg $ZnSO_4.7H_2O$, 0.003 mg $CuSO_4.5H_2O$, 0.025 mg $H_3BO_3$), supplemented with 30 g $L^{-1}$ sea salts (Sigma-Aldrich) for *L. aggregata* IAM12614 and *R. nubinhibens* ISM. All of the above defined media were additionally supplemented with 1 mM $MgSO_4$, 100 µM $CaCl_2$, and vitamins (33 µM thiamine, 41 µM biotin, 10 nM nicotinic acid). 20 mM of one of the following served as the sole source of carbon: D-glucose (Thermo Fisher), *t*3Hyp (BOC Sciences), *c*3Hyp (Chem Impex Int'l), *t*4Hyp (Bachem), *c*4Hyp (Sigma-Aldrich), or L-proline (CalBiochem).

## Plasmid construction for gene disruption

*RSP3519* was amplified from *Rhodobacter sphaeroides* 2.4.1 genomic DNA using Pfu DNA polymerase (Thermo) with primers RSP3519F and RSP3519R (*Table 9*). The resulting PCR product was inserted into the pGEM T Easy vector (Promega) to

*Table 10. Continued*

| Oligo | Sequence (5′–3′) |
|---|---|
| La28502-R | GGCACAGAAATCGAGATGCT |
| Rs16s-F | ACACTGGGACTGAGACACGG |
| Rs16s-R | TACACTCGGAATTCCACTCA |
| Rs3519-F | AGGACATCGCCTTCGAACT |
| Rs3519-R | CGATGATGCCGAAATAGTTG |
| Pa16s-F | TCACACTGGAACTGAGACACG |
| Pa16s-R | ATCAGGCTTTCGCCCATT |
| Pa1255-F | CCACCCTCTGGGAACAGTC |
| Pa1255-R | TCGTTGAGGACGAAGTTGC |
| Pa1268-F | AACAGTGGCTACCTCGGCA |
| Pa1268-R | TCGCCGACCGGTGTCTCGAT |
| Rn16s-F | ATCTGTGTGGGCGCGATT |
| Rn16s-R | GTGAGCGCATTGGTGGTCT |
| Rn08250-F | TATGGCGGCGACAGTTTC |
| Rn08250-R | GACGGCTCGAGCGTAAAC |
| Pd16s-F | GACTGAGACACGGCCCAGA |
| Pd16s-R | TCACCTCTACACTCGGAAT |
| Pd1045-F | TCGGACTACTATGTGCCGATG |
| Pd1045-R | CCTGATCGAGGCCAAAGAC |
| Pd1184-F | GCAATTTCGTGTTGAACGAG |
| Pd1184-R | CATGATGATCCAGCCCATCT |
| Pd3467-F | CTTCGCAGCCCTGTTCAT |
| Pd3467-R | GACCAGCCCTTCCTCGAT |
| Pd4859-F | GGCAAGGTGGACATCGAATA |
| Pd4859-R | CCTCGGGGTAAAGGAAGC |
| Sm16s-F | CGTGGGGAGCAAACAGGATT |
| Sm16s-R | CTAAGGGCGAGGGTTGCGCTC |
| Sm20268-F | CTGGCAAGGTGGACATCAC |
| Sm20268-R | GTAAGGCGCACTTCCTCAA |
| Sm20270-F | CGCCATGTCAATCTCCTGGT |
| Sm20270-R | GGCAGCATCCACGATCACGA |

**Table 11.** qRT-PCR primers for transcriptional analysis of genome neighborhoods

| Primer | Sequence (5′–3′) |
|---|---|
| Sliv-Sco16srRNA-F | CCGTACAATGAGCTGCGATA |
| Sliv-Sco16srRNA-R | GAACTGAGACCGGCTTTTTG |
| Sliv-Sco6289-F | GACCCTGAAGGTCGTCGTC |
| Sliv-Sco6289-R | GGTGACCGTGACGTCCAT |
| Sliv-Sco6290-F | GTCTTCTGCGGCATCGG |
| Sliv-Sco6290-R | AGTCATCGTCGTCCTCCA |
| Sliv-Sco6291-F | GCCGACCTCGACGAAGA |
| Sliv-Sco6291-R | TTGTCGGTTTCACTGCTGTC |
| Sliv-Sco6292-F | CATCGACACCAAGGTGGAC |

*Table 11. Continued on next page*

generate plasmid pRK_RSP3519-1. pRK_RSP3519-1 was digested with SmaI and ligated to a 900 bp blunt-ended chloramphenicol resistance cassette to generate pRK_RSP3519-2. pRK_RSP3519-2 was then used as the template in a PCR with primers RSP3519F and RSP3519R. The resulting product was digested with EcoRI and ligated into pSUP202 to give the plasmid used for gene disruption: pRK_RSP3519-3. To disrupt RSP3519, pRK_RSP3519-3 was electroporated into *R. sphaeroides* 2.4.1, and double crossover chromosomal gene disruptions were selected by resistance to chloramphenicol and sensitivity to ampicillin (*Matsson et al., 1998*).

## Cell preparation for gene expression analysis

Starter cultures were initiated from a single colony and grown in the appropriate rich medium overnight. This culture was used to inoculate the appropriate minimal medium (1% inoculum) supplemented with 20 mM D-glucose; the cultures were grown until $OD_{600}$ 0.3–0.5. The cells were pelleted by centrifugation (4750×$g$ for 5 min at 4°C), washed once, and resuspended in minimal medium with no carbon source. For gene expression analysis of individual PRS genes, cultures were divided into two equal volumes, 20 mM D-glucose was added to one volume and 20 mM *trans*-4-hydroxy-L-proline or *trans*-3-L-hydroxy proline was added to the other, and cultures were grown for three additional hr prior to cell harvest.

For evaluation of whole genome neighbourhoods of select PRS targets (orange, navy, hotpink, pale green, blue, and sky blue clusters) in *A. tumefaciens* C58, *B. cereus* ATCC 14579, and *S. lividans* TK24, cultures were divided into four equal volumes, supplemented with D-glucose, *trans*-4-hydroxy-L-proline, *trans*-3-hydroxy-L-proline, or L-proline to a final concentration of 20 mM, and grown until $OD_{600}$ 0.8–1.0. At the time of cell harvest, one volume of RNAprotect Bacteria Reagent (Qiagen) was added to two volumes of each culture. Samples were mixed by vortexing for 10 s and then incubated for 5 min at room temperature. Cells were pelleted by centrifugation (4750×$g$ for 5 min at 4°C), the supernatant was decanted, and cell pellets were stored at −80°C until further use.

## RNA isolation

RNA isolation was performed in an RNAse-free environment at room temperature using the RNeasy Mini Kit (Qiagen) per the manufacturer's instructions. For *B. cereus* ATCC 14579 and *S. lividans* TK24, cells were initially disrupted using a modified bead-beating procedure: cells were resuspended in 400 μl Soil Pro Lysis Buffer (MP Bio), transferred to

*Table 11. Continued*

| Primer | Sequence (5′–3′) |
| --- | --- |
| Sliv-Sco6292-R | TGACCCCGACGATGTACC |
| Sliv-Sco6293-F | GACTACGGCGTGCTCTTCAT |
| Sliv-Sco6293-R | CTCGGTGACCTCGACCAT |
| Bc0905-F | CTTCGCTGACGGACAAGTAGA |
| Bc0905-R | TGTACCGCTGTTACGGACAA |
| Bc0906-F | ACTACGAACGCAACCACACC |
| Bc0906-R | CGGAACTTGAAGGTCTCCTGT |
| Bc2832-F | TACCAGGCTTTGGTCCTGAA |
| Bc2832-R | ATTTGCCGCCAAGCTCTAAC |
| Bc2833-F | GGATGGGTTTCAGTAGCAGGA |
| Bc2833-R | CCTAGTCTTGGATAGCGAGAAGG |
| Bc2834-F | AGGTGCGTATTCGCCAGAAA |
| Bc2834-R | CCTGGCGAACGTACGATAAA |
| Bc2835-F | AACAGACCCGTGTCATCCTG |
| Bc2835-R | ACTAAGCCAGCCGGTGTATCT |
| Bc2836-F | CCTTGCATTCTCGCTTCTGT |
| Bc2836-R | AATCTTAGGAGCCCACACACC |
| Atu3947-F | TCCGGCCAAGTATGTGAAAG |
| Atu3947-R | CTATAGCCGTTCGCAGCAAG |
| Atu3948-F | ATTTCGCCCGTGATCTGTC |
| Atu3948-R | CGGCATCCACAATAATCCAG |
| Atu3949-F | GCGAACAGGCTGAAGAGATG |
| Atu3949-R | CGGCGGTAATTCCTGTTTG |
| Atu3950-F | GCTGCCGAACATATCAAGGT |
| Atu3950-R | GACCTTCGCGGTTATCTGGT |
| Atu3951-F | TGACGGACTCCAGCCTTATC |
| Atu3951-R | ATGTAACATCGGCGTGGTCT |
| Atu3952-F | GATATCGTCAAGGGCGGTTT |
| Atu3952-R | ACGCAGAGCCTTCATGTGTT |
| Atu3953-F | CAACGTCGCCAGTTACCTTC |
| Atu3953-R | GGCTGAGATCAACGACATCC |
| Atu3958-F | GGCGGCTGATACACATCTTC |
| Atu3958-R | AAAGTTGGTGCTTCGTCAGG |
| Atu3959-F | CATTCCTGACACGATCCACA |
| Atu3959-R | CAGCATCAGCAAAGGGAAGT |
| Atu3960-F | GAATGTCGTCGCCATCAAG |
| Atu3960-R | TCGTAGAGTGCCACATGCTC |
| Atu3961-F | TTCGGCACTTCTTTCTGGTC |
| Atu3961-R | GCTCGCCTGCAGATAAACA |
| Atu4675-F | TTCCTGTTATCGTCGGCACT |
| Atu4675-R | GCCTTGAAGTGAGCCTTCTG |
| Atu4676-F | ACGGCTATCGTGAAGGTCAA |
| Atu4676-R | GAATAGCTCGGGCACATCAC |
| Atu4682-F | TCCTCAGAAAGACCGACACC |
| Atu4682-R | GTGAATGTGCCGCAGGTAA |

*Table 11. Continued on next page*

Lysis Matrix E tubes (MP Bio), and agitated horizontally on a Vortex Mixer (Fisher) with Vortex Adapter (Ambion) for 10 min at speed 10. Beads and cellular debris were pelleted by centrifugation at 16,000 × $g$ for 5 min. 200 μl of the supernatant was used for subsequent RNA isolation. Cell pellets for all other organisms were disrupted according to the 'Enzymatic Lysis Protocol' in the RNAprotect Bacteria Reagent Handbook (Qiagen); lysozyme (Thermo-Pierce) was used at 15 mg ml$^{-1}$. RNA concentrations were determined by absorption at 260 nm using the Nanodrop 2000 (Thermo) and absorption ratios $A_{260}/A_{280}$ and $A_{260}/A_{230}$ were used to assess sample integrity and purity. Isolated RNA was stored at −80°C until further use.

## Reverse transcription and quantitative real-time PCR

Reverse transcription (RT) PCRs for *A. tumefaciens* C58 and *B. cereus* ATCC 14579 were performed with 300 ng of total isolated RNA using the ProtoScript First Strand cDNA Synthesis Kit (NEB) as per the manufacturer's instructions. For *S. lividans* TK24 RT-PCRs were performed with 300 ng of total RNA using the Transcriptor First Strand cDNA Synthesis Kit (Roche), with 2.5% DMSO added to relieve secondary structures. All other RT-PCRs were performed with 1 μg of total RNA using the RevertAid H Minus First Strand cDNA Synthesis Kit (Fermentas).

Primers for quantitative real-time (qRT) PCR for *A. tumefaciens* C58 and *B. cereus* ATCC 14579 gene targets were designed using the Primer3 primer tool; amplicons were 150–200 bps in length; primers for all other qRT-PCRS were designed using the Universal ProbeLibrary System (Roche); amplicons were 66–110 bps in length Primer sequences are provided in *Tables 10 and 11*. Primers were 18–27 nucleotides in length and had a theoretical $T_m$ of 55–60°C. Primer efficiency was determined to be at least 90% for each primer pair.

qRT-PCRs were carried out in 96-well plates using the Roche LightCycler 480 II instrument with the LightCycler 480 SYBR Green I Master Mix (Roche) per the manufacturer's instructions. Each 10-μl reaction contained 1 μM of each primer, 5 μl of SYBR Green I Master Mix, and an appropriate dilution of cDNA. Reactions were run as follows: one cycle at 95°C for 5 min, 45 cycles at 95°C for 10 s, 50°C for 10 s, 72°C for 10 s, and a final dissociation program at 95°C for 15 s, 60°C for 1 min, and 95°C for 15 s. Minus-RT controls were performed to verify the absence of genomic DNA in each RNA sample for each gene target analyzed. Gene expression data were expressed as crossing threshold (CT)

*Table 11. Continued*

| Primer | Sequence (5′–3′) |
| --- | --- |
| Atu4684-F | CCTCGGCAAACTCAAGGTC |
| Atu4684-R | GCGAAGAGGCAGAAGGAAA |
| Atu4691-F | AAGGGCGATATGGGTCTTTC |
| Atu4691-R | GAGCTCTTCGATGCTGTCGT |

values. Data were analyzed by the $2^{-\Delta\Delta CT}$ (Livak) method (*Livak and Schmittgen, 2001*), using the 16S rRNA gene as a reference. Each qRT-PCR was performed in triplicate, and fold-changes are the averages of at least three biological replicates.

## Data deposition

The atomic coordinates and structure factors for '4R-hydroxyproline 2-epimerases' (4HypE) from *Pseudomonas putida* F1 (citrate-liganded, PDBID:4JBD; sulfate-liganded, PDBID:4JD7), *Chromohalobacter salexigens* DSM 3043 (apo, PDBID:4JCI), *Xanthomonas campestris* (phosphate-liganded, PDBID:4JUU), *Burkholderia multivorans* (phosphate-liganded, PDBID:4K7X), *Pseudomonas fluorescens* Pf-5 (pyrrole 2-carboxylate-liganded, PDBID:4J9W; trans-4-hydroxy-L-proline-liganded, PDBID:4J9X), *Ochrobacterrium anthropic* (apo, PDBID:4K8L), and *Agrobacterium vitis* S4 (*trans*-4-hydroxy-L-proline-liganded, PDBID:4LB0) and '*trans*-3-hydroxy-L-proline dehydratase' (*t*3HypD) from *Agrobacterium vitis* S4 (pyrrole 2-carboxylate-liganded, PDBID:4K7G) have been deposited in the Protein Data Bank, www.pdb.org.

## UniProt accession IDS

This manuscript describes functional characterization of proteins with the following UniProt accession IDs: A0NXQ7, A0NXQ9, A1B0W2, A1B195, A1B196, A1B7P4, A1BBM5, A1U2K1, A3M4A9, A3PPJ8, A3QFI1, A3QH73, A3S939, A3SU01, A5VZY6, A6WW16, A6WXX7, A8H392, A9AKG8, A9AKH1, A9AL52, A9ALD3, A9AQW9, A9CFU8, A9CFU9, A9CFV0, A9CFV4, A9CFW8, A9CGZ4, A9CGZ5, A9CGZ9, A9CH01, A9CH04, A9CKB4, B0VB44, B1KJ76, B3D6W2, B4EHE6, B9J8G8, B9JHU6, B9JQV3, B9K4G4, B9R4E3, C5ZMD2, D2AV87, D2QN44, D5SQS4, D6EJK6, D6EJK7, D6EJK8, D6EJK9, D6EJL0, Q0B950, Q0B953, Q0B9R9, Q0B9S2, Q16D96, Q1QBF3, Q1QU06, Q1QV19, Q2KD13, Q2T3J4, Q2T596, Q3IWG2, Q3IZJ8, Q3JFG0, Q3JHA9, Q485R8, Q4KAT3, Q4KGT8, Q4KGU2, Q5LKW3, Q5LLV0, Q63FA5, Q6HMS8, Q6HMS9, Q73CR9, Q73CS0, Q7CFV0, Q7CTP1, Q7CTP2, Q7CTP3, Q7CTP4, Q7CTQ2, Q7CTQ3, Q7CTQ5, Q7CVK1, Q7NU77, Q81CD6, Q81CD7, Q81CD8, Q81CD9, Q81CE0, Q81HB0, Q81HB1, Q8FYS0, Q8P833, Q8YFD6, Q92WR9, Q92WS1, Q9I476, Q9I489, and Q9I492.

## Acknowledgements

This research was supported by a program project grant and three cooperative agreements from the US National Institutes of Health (P01GM071790, U54GM093342, U54GM074945, and U54GM094662). Molecular graphics and analyses were performed with the UCSF Chimera package; Chimera is developed by the Resource for Biocomputing, Visualization, and Informatics at the University of California, San Francisco (supported by NIGMS P41-GM103311). We thank Dr John 'Scooter' Morris for help in using Cytoscape for visualizing complex sequence networks. We gratefully acknowledge Rafael Toro and Rahul Bohsle for maintenance of the AECOM crystallization facility and assistance and advice on crystallization experiment assembly. Use of the Advanced Photon Source, an Office of Science User Facility operated for the U.S. Department of Energy (DOE) Office of Science by Argonne National Laboratory, was supported by the U.S. DOE under Contract No. DE-AC02-06CH11357. Use of the Lilly Research Laboratories Collaborative Access Team (LRL-CAT) beamline at Sector 31 of the Advanced Photon Source was provided by Eli Lilly Company, which operates the facility.

## Additional information

### Funding

| Funder | Grant reference number | Author |
| --- | --- | --- |
| National Institute of General Medical Sciences | P01GM071790 | Patricia C Babbitt, Steven C Almo, John A Gerlt, Matthew P Jacobson |

| Funder | Grant reference number | Author |
| --- | --- | --- |
| National Institute of General Medical Sciences | U54GM093342 | Patricia C Babbitt, Steven C Almo, John A Gerlt, Matthew P Jacobson |
| National Institute of General Medical Sciences | U54GM074945 | Steven C Almo |
| National Institute of General Medical Sciences | U54GM094662 | Steven C Almo |
| National Institute of General Medical Sciences | P41GM103311 | Steven C Almo |
| Argonne National Laboratory, Office of Science | DE-AC02-06CH11357 | Steven C Almo |

The funders had no role in study design, data collection and interpretation, or the decision to submit the work for publication.

### Author contributions

SZ, Conception and design, Acquisition of data, Analysis and interpretation of data, Drafting or revising the article; AS, Conception and design, Acquisition of data, Analysis and interpretation of data, Drafting or revising the article; XZ, Conception and design, Acquisition of data, Analysis and interpretation of data, Drafting or revising the article; MWV, Conception and design, Acquisition of data, Analysis and interpretation of data, Drafting or revising the article; RK, Conception and design, Acquisition of data, Analysis and interpretation of data, Drafting or revising the article; BH, Conception and design, Acquisition of data, Analysis and interpretation of data, Drafting or revising the article; BSF, Conception and design, Acquisition of data, Analysis and interpretation of data, Drafting or revising the article; JS, Conception and design, Acquisition of data, Analysis and interpretation of data, Drafting or revising the article; RDS, Conception and design, Acquisition of data, Analysis and interpretation of data, Drafting or revising the article; AS, Conception and design; SB, Conception and design; EA, Conception and design; AB, Conception and design; PCB, Conception and design, Drafting or revising the article; SCA, Conception and design, Analysis and interpretation of data, Drafting or revising the article; JAG, Conception and design, Analysis and interpretation of data, Drafting or revising the article; MPJ, Conception and design, Analysis and interpretation of data, Drafting or revising the article

# Additional files

## Major dataset

The following datasets were generated:

| Author(s) | Year | Dataset title | Dataset ID and/or URL | Database, license, and accessibility information |
| --- | --- | --- | --- | --- |
| Vetting MW, Toro R, Bhosle R, Al Obaidi NF, Morisco LL, Wasserman SR, Sojitra S, Washington E, Scott Glenn A, Chowdhury S, Evans B, Hammonds J, Stead M, Hillerich B, Love J, Seidel RD, Imker HJ, Gerlt JA, Almo SC | 2013 | Crystal structure of pput_1285, a putative hydroxyproline epimerase from pseudomonas putida f1 (target efi-506500), open form, space group i2, bound citrate | http://www.rcsb.org/pdb/explore/explore.do?structureId=4JBD | Publicly available at RCSB Protein Data Bank. |
| Vetting MW, Toro R, Bhosle R, Al Obaidi NF, Morisco LL, Wasserman SR, Sojitra S, Washington E, Scott Glenn A, Chowdhury S, Evans B, Hammonds J, Stead M, Hillerich B, Love J, Seidel RD, Imker HJ, Gerlt JA, Almo SC | 2013 | Crystal structure of pput_1285, a putative hydroxyproline epimerase from Pseudomonas putida f1 (target EFI-506500), open form, space group P212121, bound sulfate | http://www.rcsb.org/pdb/explore/explore.do?structureId=4JD7 | Publicly available at RCSB Protein Data Bank. |

| | | | | |
|---|---|---|---|---|
| Vetting MW, Toro R, Bhosle R, Al Obaidi NF, Morisco LL, Wasserman SR, Sojitra S, Washington E, Scott Glenn A, Chowdhury S, Evans B, Hammonds J, Stead M, Hillerich B, Love J, Seidel RD, Imker HJ, Gerlt JA, Almo SC, Enzyme Function Initiative | 2013 | Crystal structure of csal_2705, a putative hydroxyproline epimerase from CHROMOH-ALOBACTER SALEXIGENS (TARGET EFI-506486), SPACE GROUP P212121, unliganded | http://www.rcsb.org/pdb/explore/explore.do?structureId=4JCI | Publicly available at RCSB Protein Data Bank. |
| Vetting MW, Toro R, Bhosle R, Al Obaidi NF, Morisco LL, Wasserman SR, Sojitra S, Washington E, Scott Glenn A, Chowdhury S, Evans B, Hammonds J, Stead M, Hillerich B, Love J, Seidel RD, Imker HJ, Gerlt JA, Almo SC, Enzyme Function Initiative | 2013 | Crystal structure of a putative hydroxyproline epimerase from xanthomonas campestris (TARGET EFI-506516) with bound phosphate and unknown ligand | http://www.rcsb.org/pdb/explore/explore.do?structureId=4JUU | Publicly available at RCSB Protein Data Bank. |
| Vetting MW, Toro R, Bhosle R, Al Obaidi NF, Morisco LL, Wasserman SR, Sojitra S, Washington E, Scott Glenn A, Chowdhury S, Evans B, Hammonds J, Stead M, Hillerich B, Love J, Seidel RD, Imker HJ, Gerlt JA, Almo SC, Enzyme Function Initiative | 2013 | Crystal structure of a 4-hydroxyproline epimerase from burkholderia multivorans, target efi-506479, with bound phosphate, closed domains | http://www.rcsb.org/pdb/explore/explore.do?structureId=4K7X | Publicly available at RCSB Protein Data Bank. |
| Vetting MW, Toro R, Bhosle R, Al Obaidi NF, Morisco LL, Wasserman SR, Sojitra S, Washington E, Scott Glenn A, Chowdhury S, Evans B, Hammonds J, Stead M, Hillerich B, Love J, Seidel RD, Imker HJ, Gerlt JA, Almo SC, Enzyme Function Initiative | 2013 | Crystal structure of the complex of a hydroxyproline epimerase (TARGET EFI-506499, PSEUDOMONAS FLUORESCENS PF-5) with the inhibitor pyrrole-2-carboxylate | http://www.rcsb.org/pdb/explore/explore.do?structureId=4J9W | Publicly available at RCSB Protein Data Bank. |
| Vetting MW, Toro R, Bhosle R, Al Obaidi NF, Morisco LL, Wasserman SR, Sojitra S, Washington E, Scott Glenn A, Chowdhury S, Evans B, Hammonds J, Stead M, Hillerich B, Love J, Seidel RD, Imker HJ, Gerlt JA, Almo SC, Enzyme Function Initiative | 2013 | Crystal structure of the complex of a hydroxyproline epimerase (TARGET EFI-506499, PSEUDOMONAS FLUORESCENS PF-5) with trans-4-hydroxy-l-proline | http://www.rcsb.org/pdb/explore/explore.do?structureId=4J9X | Publicly available at RCSB Protein Data Bank. |
| Vetting MW, Toro R, Bhosle R, Al Obaidi NF, Morisco LL, Wasserman SR, Sojitra S, Washington E, Scott Glenn A, Chowdhury S, Evans B, Hammonds J, Stead M, Hillerich B, Love J, Seidel RD, Imker HJ, Gerlt JA, Almo SC, Enzyme Function Initiative | 2013 | Crystal structure of a putative 4-hydroxyproline epimerase/3-hydroxyproline dehydratse from the soil bacterium ochrobacterium anthropi, target efi-506495, disordered loops | http://www.rcsb.org/pdb/explore/explore.do?structureId=4K8L | Publicly available at RCSB Protein Data Bank. |
| Vetting MW, Toro R, Bhosle R, Al Obaidi NF, Morisco LL, Wasserman SR, Sojitra S, Washington E, Scott Glenn A, Chowdhury S, Evans B, Hammonds J, Stead M, Hillerich B, Love J, Seidel RD, Imker HJ, Gerlt JA, Almo SC, Enzyme Function Initiative | 2013 | Crystal structure of a 3-hydroxyproline dehydratse from agrobacterium vitis, target efi-506470, with bound pyrrole 2-carboxylate, ordered active site | http://www.rcsb.org/pdb/explore/explore.do?structureId=4K7G | Publicly available at RCSB Protein Data Bank. |

The following previously published datasets were used:

| Author(s) | Year | Dataset title | Dataset ID and/or URL | Database, license, and accessibility information |
| --- | --- | --- | --- | --- |
| Vetting MW, Toro R, Bhosle R, Al Obaidi NF, Morisco LL, Wasserman SR, Sojitra S, Washington E, Scott Glenn A, Chowdhury S, Evans B, Hammonds J, Stead M, Hillerich B, Love J, Seidel RD, Imker HJ, Gerlt JA, Almo SC, Enzyme Function Initiative | 2013 | Crystal structure of a hydroxyproline epimerase from agrobacterium vitis, target efi-506420, with bound trans-4-oh-l-proline | http://www.rcsb.org/pdb/explore/explore.do?structureId=4LB0 | Publicly available at RCSB Protein Data Bank. |
| Buschiazzo A, Goytia M, Shaeffer F, Degrave W, Shepard W, Gregoire C, Chamond N, Cosson A, Berneman A, Coatnoan N, Alzari P, Minoprio P | 2006 | Crystal structure, catalytic mechanism, and mitogenic properties of Trypanosoma cruzi proline racemase | http://www.rcsb.org/pdb/explore/explore.do?structureId=1W61 | Publicly available at RCSB Protein Data Bank. |
| Liu Y, Gorodichtchenskaia E, Skarina T, Yang C, Joachimiak A, Edwards A, Pai EF, Savchenko A, Midwest Center for Structural Genomics | 2005 | Crystal Structure of PA1268 Solved by Sulfur SAD | http://www.rcsb.org/pdb/explore/explore.do?structureId=2AZP | Publicly available at RCSB Protein Data Bank. |
| Forouhar F, Chen Y, Xiao R, Ho CK, Ma L-C, Cooper B, Acton TB, Montelione GT, Hunt JF, Tong L, Northeast Structural Genomics Consortium | 2004 | Crystal Structure of the putative proline racemase from Brucella melitensis, Northeast Structural Genomics Target LR31 | http://www.rcsb.org/pdb/explore/explore.do?structureId=1TM0 | Publicly available at RCSB Protein Data Bank. |

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
