## [Decision Letter]

Thank you for sending your work entitled “Prediction and characterization of enzymatic activities guided by sequence similarity and genome neighborhood networks” for consideration at *eLife*. Your article has been favorably evaluated by Michael Marletta (Senior editor) and 3 reviewers, one of whom, Jon Clardy, is a member of our Board of Reviewing Editors.

The Reviewing editor and the other reviewers discussed their comments before we reached this decision, and the Reviewing editor has assembled the following comments to help you prepare a revised submission.

The reviewers all thought that your contribution provides a potentially important approach to dealing with the deluge of genome sequencing data that we all face. They all felt that the neighborhood analysis – a sort of “guilt by association approach” – described a useful extension to currently used methods and that the manuscript would make a nice contribution to *eLife* with some modifications, which are outlined below. The major issues were felt to be:

1) Will the genome neighborhood approach for function assignment be generally useful with uncharacterized enzymes? The proline racemase superfamily seems like an ideal case study, since its membership can be segregated based on a limited number of (three) known activities. Thus, the annotation challenge for this enzyme family seems to be principally a choice among a small set of ‘knowns’, rather than the de novo prediction of an ‘unknown’ activity. Members of other large enzyme classes, such as metabolic kinases, phosphatases, oxidoreductases, and hydrolases, have widely varied substrates and therefore may not easily submit to functional assignment by genome neighborhood analysis. In this regard, it would be interesting to know, for a given bacterial strain, how many operons can be interrogated in a productive way by GNN analysis to yield reasonable substrate predictions for their enzyme components versus the number of operons that lack sufficient contextual annotation for substrate predictions.

2) There were instances when the description was hard to follow for a generalist reader. One example was the paragraph from the Results section, which is especially important to the general argument. In particular, what does: “We displayed the GNN as an SSN generated with an all-by-all BLAST using an e-value threshold of 10–20.” mean? Another issue with clarity was Figure 3, which seems both essential to the central argument but almost impenetrably detailed. One suggestion was to divide it into separate figures, each of which would present the minimal information needed to make one point.

For example, the authors write: “However, most clusters are multichromatic, i.e., identified by several PRS query clusters; this suggests that different query clusters in the PRS SSN have the same in vitro activity and in vivo metabolic function.” It would be very helpful, to read a detailed description of what can be learned from inspection of a single cluster illustrated in a single Figure.

---

## [Author Response]

1) Will the genome neighborhood approach for function assignment be generally useful with uncharacterized enzymes? The proline racemase superfamily seems like an ideal case study, since its membership can be segregated based on a limited number of (three) known activities. Thus, the annotation challenge for this enzyme family seems to be principally a choice among a small set of ‘knowns’, rather than the de novo prediction of an ’unknown‘ activity. Members of other large enzyme classes, such as metabolic kinases, phosphatases, oxidoreductases, and hydrolases, have widely varied substrates and therefore may not easily submit to functional assignment by genome neighborhood analysis. In this regard, it would be interesting to know, for a given bacterial strain, how many operons can be interrogated in a productive way by GNN analysis to yield reasonable substrate predictions for their enzyme components versus the number of operons that lack sufficient contextual annotation for substrate predictions.

The genome neighborhood approach for assigning functions to unknown enzymes discovered in genome projects is one of several being developed by the Enzyme Function Initiative (EFI), a large-scale multidisciplinary project supported by the National Institute of General Medical Sciences (U54GM093342; enzymefunction.org). As described in our manuscript, the proline racemase superfamily (PRS) appears to be an example of a ‘large’ functionally diverse enzyme superfamily for which all of the reactions have been described (proline racemase, 4R-hydroxyproline 2-epimerase, and *trans-*3-hydroxy-L-proline dehydratase). However, as described in the text, the PRS contains examples of convergent evolution of the same function from different ancestors; surprisingly the active site general acid/base catalysts need not be conserved to deliver the same function. Thus, despite ‘rediscovery’ of the same functions, the functions could not have been anticipated based on sequence homology; and, despite the ‘small set of knowns’ in the PRS, interesting and unexpected science was discovered.

We now have addressed the reviewers’ comments/concerns about functional assignment of large enzyme families involved in metabolic pathways in the Discussion. It is challenging to provide a meaningful estimate of the fraction of enzymes (or other proteins) in a given organism for which the Methods presented in this paper can provide correct inferences (or at least testable hypotheses). Presumably ’successes‘ would include previously characterized pathways as well as additional examples of convergent evolution of the same function. However, we have included a discussion of the occurrence of mono- and polycistronic operons in *E. coli* K-12 (40% and 60%, respectively, of the total number of genes) to acknowledge that GNNs cannot be expected to be useful for all unknown enzymes/proteins in eubacteria and archaea. We also briefly discuss the complementary use of regulon analyses to identify metabolic pathways that are not encoded by proximal operons/gene clusters.

Most importantly, we emphasize that GNNs are complementary to other approaches for function prediction. One additional source of information is protein structure, which we, and others, have used successfully to predict the substrates of enzymes, including multiple enzymes forming a pathway by metabolite docking. GNNs can be used to identify gene clusters that are most amenable to this approach, i.e., cases where multiple proteins (enzymes, transport systems, transcriptional regulators) can be homology-modeled to gain additional information about likely substrates. This structural information is particularly valuable large superfamilies of enzymes with diverse substrates, such as kinases, aldolases, etc. We also anticipate, but have not yet demonstrated, that 1) GNNs can be used to help visualize and interpret other types of information, such as high-throughput metabolomics experiments, and 2) GNNs can be constructed from interactome data rather than genome proximity, e.g., for eukaryotes.

*2) There were instances when the description was hard to follow for a generalist reader. One example was the paragraph from the Results section, which is especially important to the general argument. In particular, what does: “We displayed the GNN as an SSN generated with an all-by-all BLAST using an e-value threshold of 10-20.” mean? Another issue with clarity was*
Figure 3*, which seems both essential to the central argument but almost impenetrably detailed. One suggestion was to divide it into separate figures, each of which would present the minimal information needed to make one point.*

For example, the authors write: “However, most clusters are multichromatic, i.e., identified by several PRS query clusters; this suggests that different query clusters in the PRS SSN have the same in vitro activity and in vivo metabolic function.” It would be very helpful, to read a detailed description of what can be learned from inspection of a single cluster illustrated in a single Figure.

We thank the reviewers for pointing out text that would be difficult for the generalist. The EFI’s goal is to provide the biological community with tools and strategies for functional assignment that do not require specialized knowledge. In response to these comments, throughout the manuscript we have replaced specialized terms/concepts with explanations that should be understandable to the biological community.

With respect to Figure 3 that is essential to the central argument, we provide the ‘complete’ GNN for the PRS to illustrate its complexity in panel a. But, we also provide enlargements of specific clusters/families to illustrate/explain the importance of clusters in which all of the nodes have the same color (Figure 3) as well as clusters in which the nodes have multiple clusters (Figure 3). We discuss each of these clusters in the manuscript.

We do not believe that it is correct to present only the clusters in Figure 3; indeed, the ‘complete’ GNN contains a large amount of information about transport systems and transcriptional regulators that is not discussed in the manuscript. Detailed analyses of the clusters for transport systems and transcriptional regulators allow the conclusion that despite the conservation of enzymes in catabolic pathways, the transport systems and transcriptional regulators are not conserved. A discussion of this analysis is beyond the scope of this manuscript.